

# Modelling wind farm effects in HARMONIE-AROME (cycle 43.2.2) – part 1: Implementation and evaluation

Jana Fischereit[1], Henrik Vedel[2], Xiaoli Guo Larsén[1], Natalie E. Theeuwes[3], Gregor Giebel[1], and
Eigil Kaas[2]

[1]DTU Wind and Energy Systems, Frederiksborgvej 399, 4000 Roskilde, Denmark
[2]Danish Meteorological Institute, Lyngbyvej 100, 2100 Copenhagen, Denmark
[3]Royal Netherlands Meteorological Institute (KNMI), Utrechtseweg 297, 3731 GA De Bilt, The Netherlands

**Correspondence:** Jana Fischereit (janf@dtu.dk)

**Abstract.** With increasing number and proximity of wind farms it becomes crucial to consider wind farm effects (WFE) in the numerical weather prediction (NWP) models used to forecast power production. Furthermore, these WFE are also expected to affect other weather-related parameters at least locally. Thus, we implement the explicit wake parameterization (EWP) in the NWP model HARMONIE-AROME (hereafter HARMONIE) along-side the existing wind farm parameterization (WFP) by
Fitch et al. (2012) (FITCH). We evaluate and compare the two WFPs against research flight measurements as well as against similar simulations performed with the Weather Research and Forecasting model (WRF) using case studies. The case studies include a case for WFE above a wind farm as well as two cases for WFE at hub height in the wake of farms. The results show that EWP and FITCH have been correctly implemented in HARMONIE. For the simulated cases, EWP underestimates the WFE on wind speed and strongly underestimates the effect on turbulent kinetic energy (TKE). FITCH agrees better with
the observations and WFE on TKE are particularly well captured by HARMONIE-FITCH. After this successful evaluation, simulations with all wind turbines in Europe will be performed with HARMONIE and presented in the second part of this paper series.

## 1 Introduction

Wind turbines extract kinetic energy from the atmospheric flow to produce electricity. Thereby, they reduce the wind speed upstream, referred as blockage effect, and downstream, referred as wake effect, and sometimes increase the wind speed at the sides, referred as speed-up effect (e.g. Fischereit et al., 2022). In addition, they increase turbulence both directly through tip vortices, as well as indirectly through shear production. Hence, wind and turbulence profiles are modified around wind farms and consequently also local temperature and humidity profiles (e.g. Siedersleben et al., 2018; Baidya Roy and Traiteur, 2010).

Since wind turbines increase in number and size both on- and offshore (IRENA, 2019), their impact on numerical weather prediction (NWP) can no longer be generally ignored. According to a recent review by Fischereit et al. (2022), the Weather





Research and Forecasting (WRF) model is the most wide-spread applied model equipped with a wind farm parameterization (WFP). Extensive validation of WRF with the built-in WFP by Fitch et al. (2012), hereafter 'FITCH', has been performed as summarized in Fischereit et al. (2022). Beside WRF+FITCH, the Explicit Wake Parameterisation (Volker et al., 2015), hereafter EWP, is the second most frequently applied WFP in WRF according to the review in Fischereit et al. (2022). In addition, FITCH has been implemented in other NWP models, among others in HARMONIE-AROME (Bengtsson et al., 2017, hirlam.org), hereafter HARMONIE, by van Stratum et al. (2022).

While WRF+WFP has been extensively applied and verified as summarized in Fischereit et al. (2022), WFPs in HARMONIE are still relatively unexplored. Since HARMONIE is used by at least 11 national weather services in Europe, it is relevant to also integrate WFE in HARMONIE. van Stratum et al. (2022) started this process with the implementation of FITCH. They evaluated a one-year long simulation against measurements of power production, from lidar and mast, as well as in a case study against aircraft measurements. They showed that using FITCH provided a more realistic representation of the atmosphere near wind farms than a simulation without WFP. In this study, we extend the work by van Stratum et al. (2022) by implementing the EWP into HARMONIE. Having two WFPs available is advantageous, because it allows to create an ensemble of possible wind farm effects (WFE), highlighting the uncertainty of the forecast.

The work is divided into two parts. Part 1 is presented in this manuscript and describes the implementation of EWP in HARMONIE as well as the comparison against WRF results and flight measurements for three case studies. Part 2, presented in Fischereit et al. (2023a), deals with the setup of a wind turbine database for Europe, long-term evaluation of the HARMONIE simulations as well as the sensitivity of the forecast to the applied WFP.

Part 1 in the present manuscript is structured as follows: key features of EWP and the implementation of EWP into HARMONIE are described in Sect. 2.1. The model set-up for HARMONIE and WRF for the simulations are described in Sect. 2 and the results and comparisons are shown in Sect. 3, and discussed in Sect. 4 and concluded in Sect. 5.

## 2 Modelling frameworks and case studies

In the following, the applied WFPs, models and model set-ups are described. Sect. 2.1 describes the characteristics of the EWP and highlights the differences to the more widely used FITCH (Fitch et al., 2012). In Sect. 2.2 and Sect. 2.3 introduces the two applied NWP models, HARMONIE and WRF, respectively. The investigated case studies are described in Sect. 2.4 and Sect. 2.5.

### 2.1 Implementation of EWP in HARMONIE

To parameterize the effect of wind farms on the atmosphere EWP imposes an elevated momentum sink or drag force on a control volume $\Delta V$ of the flow $U$, which acts on the rotor area $A_r$ and is proportional to the thrust coefficient $C_T$:

$$\overline{f}_t = \frac{1}{2} C_T U^2 A_r / \Delta V. \tag{1}$$




This is similar to the FITCH parameterization. However, in contrast to FITCH, EWP accounts for a sub-grid scale vertical wake expansion. The idea behind this is that due to the size of the mesoscale grid cell of typically 1–5 km$^2$, the wind turbine wake has expanded vertically when reaching the grid cell boundary. To account for this effect, EWP builds upon classical wind
turbine wake theory, by assuming an exponential expansion based on an effective length scale $\sigma_e$:

$$f_{xyz} = N_t \sqrt{\frac{\pi}{8}} \frac{C_T r_h^2 |\overline{u}_{rh_{xy}}|^2}{\Delta x \Delta y \sigma_e} \exp \left[ \frac{1}{2} \left( \frac{z_z - h}{\sigma_e} \right)^2 \right] \tag{2}$$

Here, $N_t$ is the number of turbines per grid cell, $r_h$ is rotor radius $= 0.5d$ with $d$ being the rotor diameter, $z_z$ is height of model level $z$, $h$ is hub height and the thrust coefficient is used as a function of wind speed at hub height $C_T = C_T(\overline{u}_{rh})$. The effective length scale, $\sigma_e$, is related to the model grid size ($L = 0.5\Delta x$), the turbulent diffusion coefficient from mesoscale
turbulence scheme ($K$) and an initial length scale that represents the unresolved wake expansion in the near wake $\sigma_h = f_r r_h$ with $f_r$ being a tuneable wake expansion scaling factor:

$$\sigma_e = \frac{\overline{u}_{rh}}{3KL} \left[ \left( \frac{2K}{\overline{u}_{rh}} L + \sigma_h^2 \right)^{3/2} - \sigma_h^3 \right] \tag{3}$$

Using the wind direction, $WD$, the drag force $f_{xyz}$ is split into the two wind components:

$$\overline{f}_{1,xyz} = f_{xyz} \cos(WD_{xyz}) \quad \text{and} \quad \overline{f}_{2,xyz} = f_{xyz} \sin(WD_{xyz}) \tag{4}$$

Another difference between FITCH and EWP is the treatment of turbulent kinetic energy (TKE) within these schemes. For EWP, Volker et al. (2015) assumes that the heterogeneous part of the mean flow (e.g. organized motions) is part of the mean flow kinetic energy and not part of random TKE. Based on that, the remaining addition of TKE due the rotation of wind turbines is negligible in a mesoscale model. In FITCH such a distinction is not made and thus an explicit TKE source term is added to the mesoscale model equations (Fitch et al., 2012). In both the EWP and the FITCH scheme, wind turbines are an
implicit source of TKE through shear generated turbulence arising from the turbine wake. More details on the derivation for EWP are given in Volker et al. (2015) and more discussions on the difference between FITCH and EWP can be found in e.g. Fischereit et al. (2022).

The implementation of EWP in HARMONIE follows the implementation of FITCH as described in van Stratum et al. (2022). The only difference is that for EWP the TKE tendencies are not modified and a different drag force is used. The
75 turbulent diffusion coefficient, $K$, is used to calculate the wake expansion. $K$ is derived from the stability corrected turbulence length scale $\ell$ (Lenderink and Holtslag, 2004) and TKE from the planetary boundary layer scheme in HARMONIE:

$$K = \ell \cdot \sqrt{\text{TKE}} \tag{5}$$

For the implementation of Eq. 4 not the true wind direction $WD$ is used, since $u$ and $v$ are grid-following in HARMONIE and thus are not necessarily aligned north-south and east-west.





## 2.2 HARMONIE

We implement EWP in HARMONIE-AROME cycle 43.2.2. HARMONIE is a non-hydrostatic, convection permitting limited-area NWP model that is developed within the HIRLAM-C consortium (Bengtsson et al., 2017, hirlam.org). The dynamics are based on the fully compressible Euler equations (Simmons and Burridge, 1981), which are solved numerically using a semi-Lagrangian advection scheme with semi-implicit time stepping (Bénard et al., 2010). The physical parameterizations include a

85 multiband radiation scheme, prognostic equations for liquid and solid hydrometeors, a prognostic equation for turbulent kinetic energy and a mass-flux based shallow convection scheme called EDMFm. Surface physics is modelled using the SURFEX scheme (Masson et al., 2013). For further details on HARMONIE see Bengtsson et al. (2017).

Here we use the model grid design of the operational NWP setup at the Danish Meteorological Institute (DMI): We simulate the Northern Europe DMI domain NEA, which covers all of Scandinavia, UK, Iceland and parts of Germany, and is centered

around 60°N and 7°E (Fig. 1b). It has a horizontal resolution of 2.5 km and 65 vertical layers, running from the surface up to 10 hPa. The lowest 10 full levels are most relevant for this study, located approximately at 12 m, 38 m, 63 m, 89 m, 117 m, 146 m, 177 m, 211 m, 247 m, 287 m height above the ground. These levels along with those defined in WRF and rotor areas for the different turbine types are shown in Fig. 2.

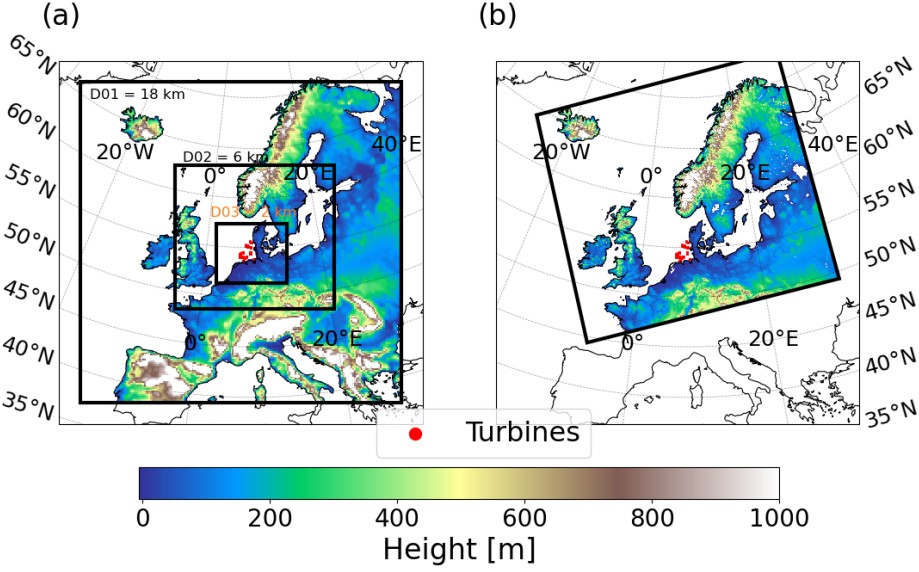

**Figure 1.** (a) Nested WRF domain and (b) HARMONIE domain 'NEA' and employed in this study.

We run HARMONIE in forecasting mode and use hourly IFS fields from ECMWFs global forecast model as lateral boundary

conditions. A warm-up period of 7 days prior to the case study days (Sect. 2.4) is used to spin-up the simulations. The long spin-up period is needed due to advanced 3D-VAR data assimilation used in HARMONIE. After the spin-up period using 3-hour cycling, 24-hour forecasts are made at 00 and 12 UTC. The data assimilation includes surface synoptic observation (SYNOP)

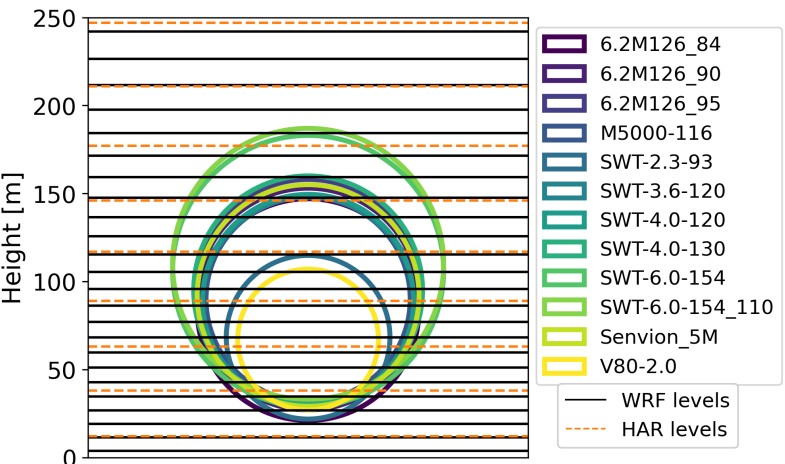

**Figure 2.** WRF (black solid) and HARMONIE (orange dashed) model levels and rotor areas of the different turbine types used in this study.

pressures, radiosonde winds, temperatures and humidities, buoy pressures, aircraft winds and temperatures (AMDAR, AIREP, ACRAS, MODE-S). In addition several types of surface and near surface data are assimilated. For sea surface temperatures

OSTIA (Donlon et al., 2012) is used. The evaluated forecast was performed at the closest synoptic hour, which corresponds to 12:00 for the three test cases. We use the parameterizations and settings mostly corresponding to the operational forecasts at DMI. The only exception is the use of WFPs and the number and type of assimilated observations. A summary of all settings is given in Table 1.

## 2.3    WRF

We apply WRF 4.2.2 (Skamarock et al., 2019) in a nested domain set-up with the corresponding spatial resolutions 18 km, 6 km and 2 km shown in Fig. 1a. We use 80 non-equidistant vertical model levels with more levels in the lowest 200 m of the atmosphere around the rotor (Fig. 2, Table 1). We run WRF in hindcast mode and use ERA5 data as the initial and boundary conditions. More details on the physical schemes are given in Table 1. The settings follow mostly those in Larsén and Fischereit (2021), since in that study good agreement for the mean wind structures was found between the simulations and three types of

measurements.

All WRF simulations are initialized at midnight and run for 24 hours, except for the simulation on 15 October 2017. For the simulation on 15 October 2017, it was investigated whether initialization at 00:00 or 06:00 would show better results. The simulation initialized on 00:00 underestimated the wind speed and therefore we used 06:00 as the initial time. The later initialization causes the simulations follow more closely the boundary conditions, which has an advantage in changing meteo-

rological situations. A similar behaviour was reported in Larsén and Fischereit (2021) for 14 October 2017 and could be solved by initializing the simulations at 06:00. Since the analysis presented in this study starts after 12:00 a spin-up time of 6 hours is still maintained. The other simulations use a spin-up time of around 12 hours.



**Table 1.** Model settings for HARMONIE and WRF.

| Parameter | HARMONIE | WRF |
|---|---|---|
| **Version** | 43.2.2 with Fitch and EWP implemented | 4.2.2 with EWP implemented |
| **Spatial settings** | | |
| Domain (Fig. 1) | 2.5 km uniformly (1200 x 1080 pts) | One-way nested: D1: 18 km (202 x 202 pts), D2: 6 km (301 x 271 pts), D3: 2 km (394 x 334 pts) |
| Center location | 60°N and 7°E | 55.5°N and 6°E |
| Vertical levels (Fig. 2) | 65 levels, $\Delta z \approx 26$ m up to 117 m | 80 levels, $\Delta z \approx 10$–25 m up to 200 m |
| **Temporal settings** | | |
| Simulation length | 7 days (8 days for 15 October 2017) | 24 hours |
| Spin-up | 7.5 days | 12 hours (6 hours for case study 15 October 2017) |
| Update interval | Data assimilation every 3 hours, 24 h forecasts at 00:00 and 12:00 | Boundary conditions every 6 hours, spectral nudging applied above the boundary layer |
| Output interval | 15 min | 10 min |
| **Initialization and boundary** | | |
| Forcing data | ECMWF daily forecasts (hourly, 18 km horizontal, 137 vertical levels) | Reanalysis: ERA5 (Hersbach et al., 2018) on pressure levels |
| Terrain data | Combination of local high resolution datasets | GMTED2010 (Danielson and Gesch, 2011) |
| Land use data | Combination of local high resolution datasets | ESA CCI[1] |
| Sea surface temperature | OSTIA (Donlon et al., 2012) | OSTIA (Donlon et al., 2012) |
| **Physics scheme** | | |
| Microphysics | ICE-3: Pinty and Jabouille (1998), Lascaux et al. (2006), Bouteloup et al. (2011), Bengtsson et al. (2017) | Thompson et al. (2008) (option 8) |
| Radiation | Long-wave: 16 band RRTM. Short-wave: Bengtsson et al. (2017) | RRTMG Iacono et al. (2008) (option 4) |
| Cumulus | Deep convection resolved. Shallow convection: EDMFm (Neggers et al., 2007; Siebesma et al., 2007; de Rooy and Siebesma, 2008; de Rooy and Pier Siebesma, 2010) | For D1 only: Kain (2004) (option 1) |
| Land surface | SURFEX (Masson et al., 2013) | Noah LSM (Tewari et al., 2004) (option 2) |
| Planetary boundary layer | Turbulence parameterisation with HARATU (de Rooy et al., 2022) | MYNN2.5 (Nakanishi and Niino, 2009) (option 5) |
| Wind farm parameterization | EWP with $f_r = 1.7$ | EWP with $f_r = 1.7$ |
| | FITCH with $f_{TKE} = 1$ | FITCH with $f_{TKE} = 1$ |

[1] Available from https://www.esa-landcover-cci.org/





## 2.4 Case studies

We investigate three case studies (Table 2) to evaluate the implementation of EWP in HARMONIE. One of the main reasons
for choosing the three cases is that open-access high-resolution flight measurements conducted within the German WIPAFF project (Bärfuss et al., 2019) are available. These cases also include a variety of conditions of wind speed, wind direction and stability, which enable us to evaluate the WFP performance in different background conditions. In addition, the cases differ in the type of WFE that was measured: within the wake at around hub height or above the wind farm. Thus, by choosing these cases the implemented WFP can be evaluated for different effects. This also extends the evaluation performed in van
Stratum et al. (2022), which only included one evaluation within the wake at hub height. In addition, we here focus also on other parameters, especially TKE, which was not included in the previous evaluation.

The flight tracks during the three cases in relation to the corresponding wind farms are shown in Fig. 3. The case study on 14 October 2017 (blue lines in Fig. 3) is used to evaluate the performance of the simulations above a wind farm. This case follows up on existing analysis in Larsén and Fischereit (2021) and Siedersleben et al. (2020). The other two cases evaluate the
130 wake of the farm around hub-height (orange and purple lines in Fig. 3). Background and evaluation objective are summarized in Table 2.

**Table 2.** Investigated case studies. Background conditions refer to the first transect upwind of the farm.

| Date | Wind farms | Flight height (mean height) | Background conditions | Evaluation objective |
|---|---|---|---|---|
| 08 August 2017 | Amrumbank West and Nordsee Ost | Wake at hub height (91 m) | 80° (E); 13 ms$^{-1}$ | wake with temperature effect |
| 14 October 2017 | Gode Wind 1+2 and Nordsee One | Above wind farm (250 m) | 250° (W-SW); 15 ms$^{-1}$ | effects above the farm |
| 15 October 2017 | Gode Wind 1+2 and Nordsee One | Wake slightly above hub height (122 m) | 190° (S-SW); 11 ms$^{-1}$ | wake without temperature effect |

The raw flight data was divided into transects, which are shown as darker stretches in Fig. 3. Those transects were approximately perpendicular to the mean upwind wind direction. The mean flight height for the three case studies were 91 m, 122 m and 250 m. The transect flight data are sampled at a frequency of 100 Hz with the aircraft ground speed being 66 m s$^{-1}$ (Platis
et al., 2018), which corresponds to a spatial resolution of 0.66 m. The data include among others the three wind components ($u$, $v$ and $w$), temperature and humidity. We calculate TKE using the standard deviation, $\sigma$, of the three wind components: $TKE = 0.5 \cdot (\sigma_u^2 + \sigma_v^2 + \sigma_w^2)$. For the calculating the standard deviations, we use the same data window of 2 km, as in Larsén and Fischereit (2021) and close to that used in Platis et al. (2020).

Besides the WIPAFF measurements, we also use synthetic aperture radar (SAR) data taken from https://science.globalwindatlas.
info/#/map (Badger et al., 2022) to evaluate the general meteorological situation during the case studies. The SAR images are retrieved from ENVISAT and combined with an empirical transfer function to derive the neutral 10 m wind speed from the radar backscatter of small locally generated waves. All shown SAR images in Sect. 3 are made up of more than one SAR scene

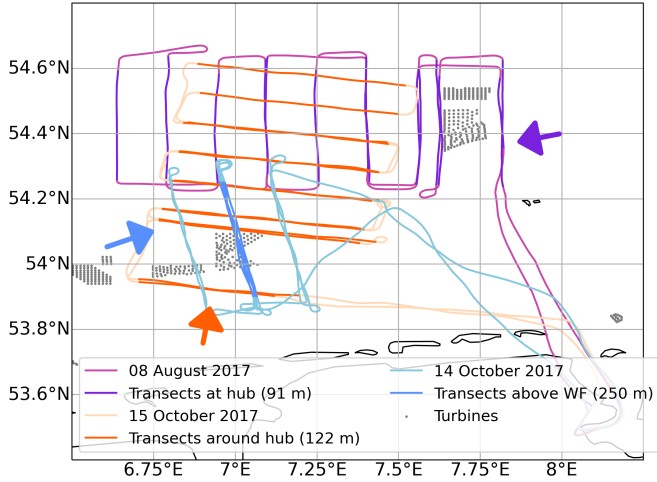

**Figure 3.** Flight tracks (brighter lines) and focus transects (darker lines) for the three case studies 08 August 2017 (purple), 14 October 2017 (blue), and 15 October 2017 (orange) along with the turbines of interest in that area (gray) and the mean wind speed and direction for the cases (colored arrows) derived from the first upwind transects with respect to the farm (Table 2).

recorded at around 17:00 UTC. The area in the SAR scene does not always cover the full flight track, but serves to assess the background conditions.

## 2.5 Wind turbines

The position and types of wind turbines are taken from Larsén and Fischereit (2021) for both the HARMONIE and WRF simulations. The rotor areas with respect to the vertical grid for the various turbines are shown in Fig. 2 and the positions are shown in Fig. 1, respectively. Since the flight measurements (Sect. 2.4) were conducted in 2017, we only include wind turbines present in the North Sea at that time. For the case studies, we focus on four wind farms (Table 2): for the case studies on 14.08.2017 and 15.08.2017 on Gode Wind 1+2 and Nordsee One, and for the case study on 08 August 2017 on Amrumbank West and Nordsee Ost. Gode Wind 1+2 and Nordsee One are equipped respectively with 110 m tall SWT-6.0-154 wind turbines (SWT-6.0-154_110 in Fig. 2) and 90 m tall 6.2M126 (6.2M126_90 in Fig. 2). In Amrumbank West and Nordsee Ost 88 m tall SWT-3.6-120 wind turbines and 95 m tall 6.2M126 wind turbines are installed, respectively. The thrust and power curves of those three wind turbine models are shown in Fig. 4.

The turbines are assigned to the grid cells in WRF and HARMONIE. Figure 5 shows, how many turbines are assigned to the each grid cell around the wind farms of interest.

We simulate three scenarios for each case study (Table 2) for both the HARMONIE and WRF simulations. The three scenarios are (1) a scenario without WFEs included (denoted 'NWF'), (2) a scenario with the parameterization by Fitch et al. (2012) (denoted 'FITCH') and (3) a scenario with the EWP parameterization (denoted 'EWP'). For FITCH we use a turbulent kinetic energy (TKE) factor of 1 in WRF to make it comparable to the implementation in HARMONIE, which does not include



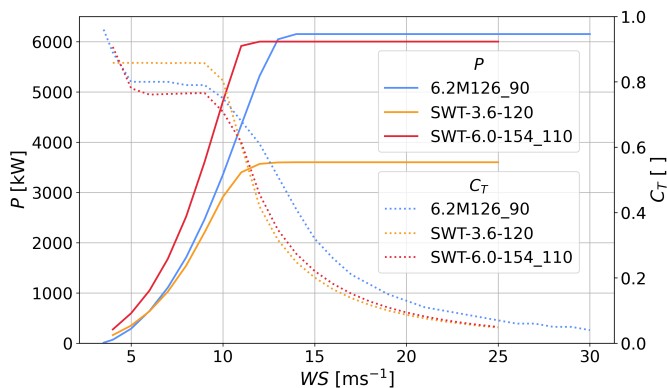

**Figure 4.** Thrust and power curves for the turbine types of the four wind farms of interest. Note that the power and thrust curves for 6.2M126 are identical for the 90 m and 95 m version of the turbine.

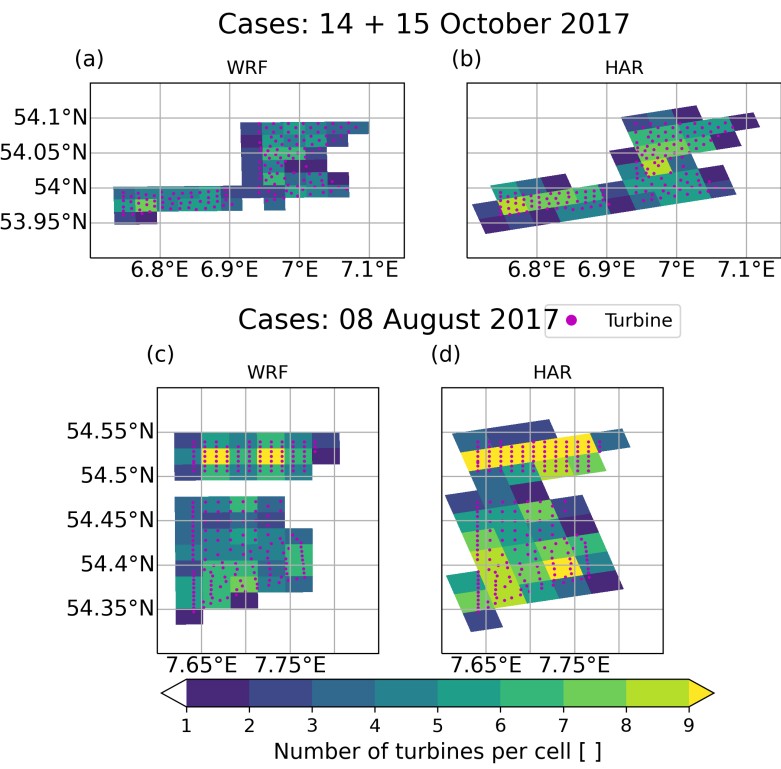

**Figure 5.** Number of turbines per grid cell in (a,c) WRF and (b,d) HARMONIE for (a,b) the wind farms relevant for case study 14. and 15 October 2017 and (c,d) the winf farms relevant for case study 08 August 2017.

a TKE factor. The correction factor for TKE was introduced following the discovery of a bug in the implementation of turbine-generated TKE in the WRF model (Archer et al., 2020). It is an empirical factor that was derived based on the best agreement





between a LES simulation and a WRF-FITCH simulation. Using this comparison Archer et al. (2020) suggested to use a factor of 0.25. However, a subsequent study by Larsén and Fischereit (2021) found inconclusive results when comparing simulations
using correction factors of 0.25 and 1 with measurements. Thus, while a correction factor of 1 deviates from the default, the 'best' choice is still unclear. Hence, it is reasonable to use a correction factor of 1, i.e. no correction factor, here. For EWP we set the tuneable initial wake expansion coefficient, as introduced in Eq. 3, to $f_r$ =1.7. Previous studies (e.g. Volker et al., 2015; Larsén and Fischereit, 2021) found that simulation results are not very sensitive to variation in $f_r$ between 1.5 and 1.7.

### 2.6 Evaluation metrics

To assess the agreement between observations and simulations, several error measures are used to evaluate the different components of the overall error: the bias (BIAS) assesses the systematic error, the standard deviation of errors (STDE) assesses the non-systematic error and the root mean square error (RMSE) assesses the combined error. In addition, the correlation coefficient is derived to assess the temporal agreement with the observations. The equations for the different error measures can be found for instance in Schlünzen and Sokhi (2008). The error metrics are calculated for each transect and the median error over
all transects is given in this manuscript.

Besides the agreement with measurements, the magnitude of the wind farm effect is evaluated. To characterize the magnitude, the difference between simulations with wind farms (FITCH / EWP) and simulations without wind farms (NWF) are calculated: FITCH-NWF and EWP-NWF for both WRF and HARMONIE. The correctness of the magnitude of the WFE cannot be simply assessed against the flight observations, since there are no observations without WFE. To circumvent this problem, artificial
observations without WFE ('obsNWF') are constructed based on a simple linear interpolation between two locations at either sides of the farm (or wake) on a flight transect. Those artificial observations are shown exemplary as gray dash-dotted lines in Fig. 6. Since the background conditions also vary in time and space and the width of the wake increases with increasing distance from the farm, it is difficult to define which part of the track is already under wake influence and which part is not. Thus, two different locations are chosen to measure the uncertainty of the observational wake effect. The two locations are
based on the location of the minimum and maximum $WS$ value to the left of the farm (or wake) with respect to the mean wind direction (Fig. 6, left red crosses).

The presented method is very simplistic, but provides a way to quantify the wake effect in the observations. The method is similar to the method presented in Cañadillas et al. (2020), but instead of evaluating the maximal WFE it provides the mean WFE. In addition, it also works for the case of measurements above the farm and for other variables than $WS$. As a
comparison also the method by Cañadillas et al. (2020) is applied for the wake cases. Cañadillas et al. (2020) use an exponential function $U_R(x) = 1 - a \cdot \exp(-bx)$ for the wind speed recovery $U_R$ as function of downstream distance $x$ with coefficients $a$ and $b$ given in their study. The reference wind speed $U_{ref}$ for each transect to calculate the maximal WFE at each $x$ as $\mathrm{WFE}(x) = U_{ref}(x) - U_R(x) \cdot U_{ref}(x)$ is not provided in Cañadillas et al. (2020). Therefore, the reference wind speed is derived as mean over the three points of 'obsNWF' (Fig. 6, red crosses).



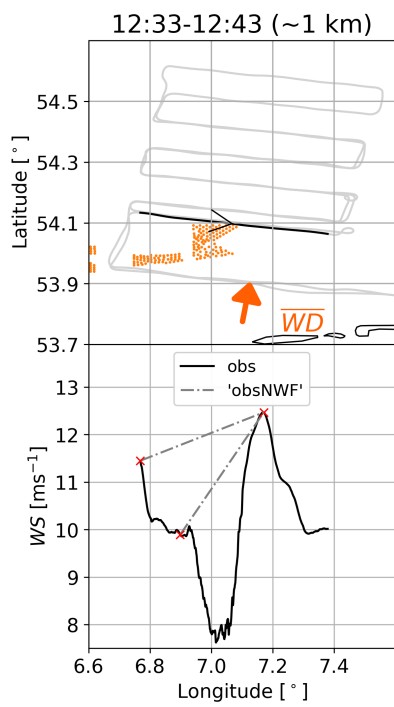

**Figure 6.** First row: Flight track (gray) with transect of interest (black) and turbines (orange); second row: wind speed ($WS$) for flight measurements (obs, black, solid) for the case 15 October 2017. Observational wake effects (obsNWF, gray) are as shown as dashed-dotted lines. Mean upwind transect wind direction ($\overline{WD}$, as in Table 2) is shown by the arrow. The title indicates the time of the transect and the median downstream distance from the farm.

**3  Results**



## 3.1 Above a wind farm

### 3.1.1 Background conditions

In this case study, WRF and HARMONIE simulations with and without WFP are evaluated with WIPAFF aircraft measurements above the wind farm on 14 October 2017 (Table 2). During this case, the wind was from south-west (Table 2, Fig. 3), the
atmosphere was stably stratified and low-level jets were present as discussed in detail in Siedersleben et al. (2020) and Larsén and Fischereit (2021). The prominent wind direction can also be seen in the wind farm wake direction visible in the 10 m winds from SAR and simulations (Fig. 7). HARMONIE winds simulated for 10 m height agree visually well with the SAR-derived winds, while WRF slightly overestimates the wind. Both models and both parameterizations, i.e. FITCH and EWP, visually capture extend of the farm wakes well compared to SAR.

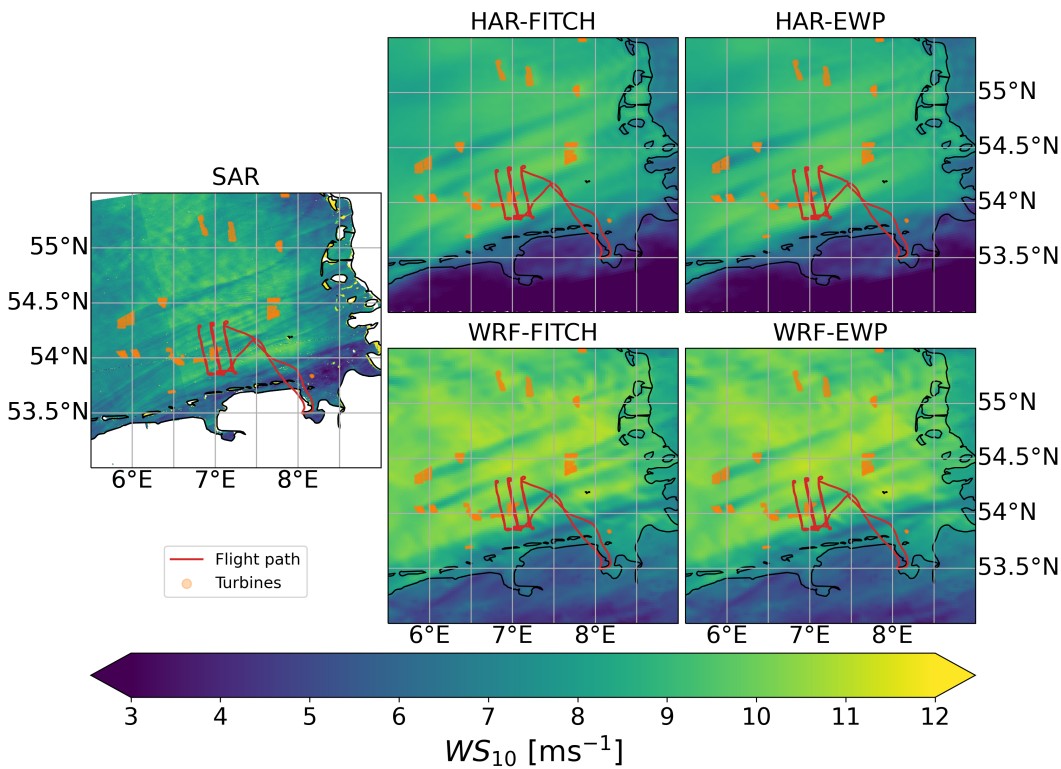

**Figure 7.** Wind speed at 10 m height from SAR on 14 October 2017 at 17:17, HARMONIE (HAR) with FITCH and EWP WFPs at 17:15 and WRF with FITCH and EWP WFPs at 17:20. The red lines are the flight path.





### 3.1.2 Transects above the farm

The measurement aircraft flew at a height of about 250 m six times above Gode Wind 1+2, as can be seen in the flight track in Fig. 8 (upper row). The corresponding time is given in the title of sub-figures in Fig. 8. While all transects were measured within 2 hours, wind speed varies quite considerably (Fig. 8, center row) by 1 ms$^{-1}$. Neither WRF nor HARMONIE can fully capture this temporal variability. Although WRF overestimated the 10 m wind speed if compared to SAR (Fig. 7), the wind speed at 250 m matches well with the observations for some transects as well as with HARMONIE.

Above Gode Wind 1+2 $WS$ is decreased and TKE increased (Fig. 8, second and third row) due to effect of the wind farm. Both WRF and HARMONIE equipped with the FITCH WFP (blue) can capture this effect. The EWP (orange and red) can capture the wind speed reduction, but underestimates the magnitude of the reduction in both models. Furthermore, EWP does not capture the increased magnitude of TKE above the farm. However, in contrast to the simulation without wind farms (NWF, yellow), EWP shows a reduced $WS$ above the farm. Due to the relatively coarse resolution of 2 km in WRF and 2.5 km in HARMONIE, the speed-up at the side of the farm cannot be properly captured.

The wind direction, $WD$, is slightly off in HARMONIE (Fig. 8 bottom row), especially in the earlier transects up to 15:50. As a consequence also the exact location of the wind speed deficit is not as well captured in HARMONIE as it is in WRF (Fig. 8 second row).

### 3.1.3 Error statistics

The quantitative evaluation of the simulations against the flight measurements is difficult, since for some transects the background wind is not well simulated. This is visible at both side of the farms, where HARMONIE and WRF overestimate $WS$ (Fig. 8 second row). Since the thrust coefficient depends on wind speed (see SWT-6.0-154-curve in Fig. 4), the WFE differs for different background conditions. This is a common problem for evaluating WFPs as found in the review in Fischereit et al. (2022).

In our case, the wind speed is within the range of 14 ms$^{-1}$–16 ms$^{-1}$, where the sensitivity of the thrust is already reduced compared to lower wind speeds, but still high (Fig. 4). To circumvent this problem, we calculate several error metrics that assess the different components of the overall error as described in Sect. 2.6. All error metrics are calculated both for $WS$ and TKE, as well as for the two models with each 3 different scenarios as shown in Sect. 2.5.

The statistics (Table 3) confirm that FITCH agrees best with the observations and that EWP performs reasonable for $WS$, but as bad as NWF for TKE. Overall the error measures indicate comparable performance of WRF+WFP and HARMONIE+WFP, which indicates that that the implementation of EWP was successful.

### 3.1.4 Wind farm effects

To evaluate the WFE above the farm (60 m above the rotor tip), the NWF-scenario is subtracted from the simulations with wind farms as described in Sect. 2.6. Those differences are shown in Table 4. It shows that wind speed deficits at that height above the farm are around -0.75 ms$^{-1}$ and TKE increase is around 0.5 m$^2$s$^{-2}$ according to the observations. However, there

**Figure 8.** First row: Flight track (gray) with transect of interest (black) and turbines (orange) and mean upwind transect wind direction ($\overline{WD}$, as in Table 2) as arrow with respect to flight track; second row: wind speed ($WS$); third row: turbulent kinetic energy (TKE); fourth row: wind direction ($WD$) for flight measurements (black), WRF simulations (brighter colours, densely broken lines) and HARMONIE simulations (darker colours, loosely broken lines) for the no-wind-farm scenario (NWF, yellow, solid and dashed dotted lines), EWP parameterization (red, dotted lines) and FITCH parameterization (blue, dashed lines) are shown for the median flight height of each transect (around 250 m). For each simulation two lines for the nearest model output time steps with shaded area between them are shown. Observational wake effects (obsNWF) are as shown as gray dashed-dotted lines (Sect. 2.6). The title of each column corresponds to the time of the respective transect. All transects are at a height of around 250 m.

is some uncertainty in the artificially generated NWF observations. FITCH matches the magnitudes quit well, although it underestimates the mean TKE effect and slightly overestimates the $WS$ effect. EWP slightly underestimates the WFE with respect to $WS$ and has almost no WFE with respect to TKE.

Comparing the results for HARMONIE and WRF shows that mean WFE are slightly higher for HARMONIE compared to WRF for both FITCH and EWP (Table 4). There could be multiple reasons for this. Firstly, the different planetary boundary





**Table 3.** Median error metrics over all transects for WRF (three left-most columns) and HARMONIE (three right-most columns) for three scenarios each: FITCH and EWP WFP and no-wind-farm (NWF) scenario. The error metrics bias (BIAS), standard deviation of errors (STDE), the root mean square error (RMSE) and the correlation coefficient are shown for both wind speed ($WS$) and turbulent kinetic energy (TKE). The cells are color coded row by row with respect to performance, with light color indicating best performance.

| | | WRF | | | HARMONIE | | |
| | | FITCH-obs | EWP-obs | NWF-obs | FITCH-obs | EWP-obs | NWF-obs |
|---|---|---|---|---|---|---|---|
| $WS$ | BIAS [ms$^{-1}$] | 0.54 | 1.07 | 1.40 | 0.72 | 1.17 | 1.69 |
| | CORR | 0.90 | 0.90 | 0.59 | 0.90 | 0.82 | 0.21 |
| | RMSE [ms$^{-1}$] | 0.62 | 1.21 | 1.63 | 0.88 | 1.27 | 1.87 |
| | STDE [ms$^{-1}$] | 0.40 | 0.53 | 0.83 | 0.43 | 0.51 | 0.88 |
| TKE | BIAS [m$^2$ s$^{-2}$] | -0.11 | -0.35 | -0.36 | -0.06 | -0.34 | -0.37 |
| | CORR | 0.84 | -0.12 | -0.24 | 0.86 | -0.04 | -0.23 |
| | RMSE [m$^2$ s$^{-2}$] | 0.32 | 0.62 | 0.62 | 0.29 | 0.64 | 0.66 |
| | STDE [m$^2$ s$^{-2}$] | 0.30 | 0.51 | 0.52 | 0.29 | 0.55 | 0.55 |

layer schemes applied in WRF and HARMONIE (Table 1). Secondly, the horizontal and vertical resolution differ between WRF and HARMONIE (Table 1) and consequently the wind turbines are differently assigned to the grid cells (Fig. 5).

**Table 4.** Median wake effect over all transects for WRF (two left-most columns), HARMONIE (two center columns) and observations (right-most column) for the difference of FITCH and no-wind-farm (NWF), EWP and NWF scenario and observations (obs) and artifically generated NWF-observations for both wind speed ($WS$) and turbulent kinetic energy (TKE).

| | WRF | | HARMONIE | | Observations |
| | FITCH-NWF | EWP-NWF | FITCH-NWF | EWP-NWF | obs-obsNWF |
|---|---|---|---|---|---|
| $WS$ [ms$^{-1}$] | -0.91 | -0.38 | -1.01 | -0.56 | -0.73±0.10 |
| TKE [m$^2$ s$^{-2}$] | 0.26 | 0.01 | 0.32 | 0.02 | 0.54±0.00 |

## 3.2 In the wake of a wind farm

To evaluate the model performance in the wake of the wind farm at around hub height, we look at two case studies: 15 October 2017 and 08 August 2017. The case 15 October 2017 is chosen because the background meteorology was better matched from the simulations due to a less complex meteorological situation. The case 08 August 2017 is chosen because an effect on the temperature from the wind farm was observed, which is interesting to consider from a NWP point of view, which does not focus solely on power forecast.





### 3.2.1 Background conditions

On 15 October 2017 around the flight time wind was coming from the southern coast of the German Bight. The aircraft was flying at about 120 m height in the wake of Nordsee One and Gode Wind 1+2. The atmosphere was slightly stable stratified (Cañadillas et al., 2020). Compared to SAR, both HARMONIE and WRF overestimate the 10 m wind speed, but show the same gradient of increasing wind towards the North (Fig. 9).

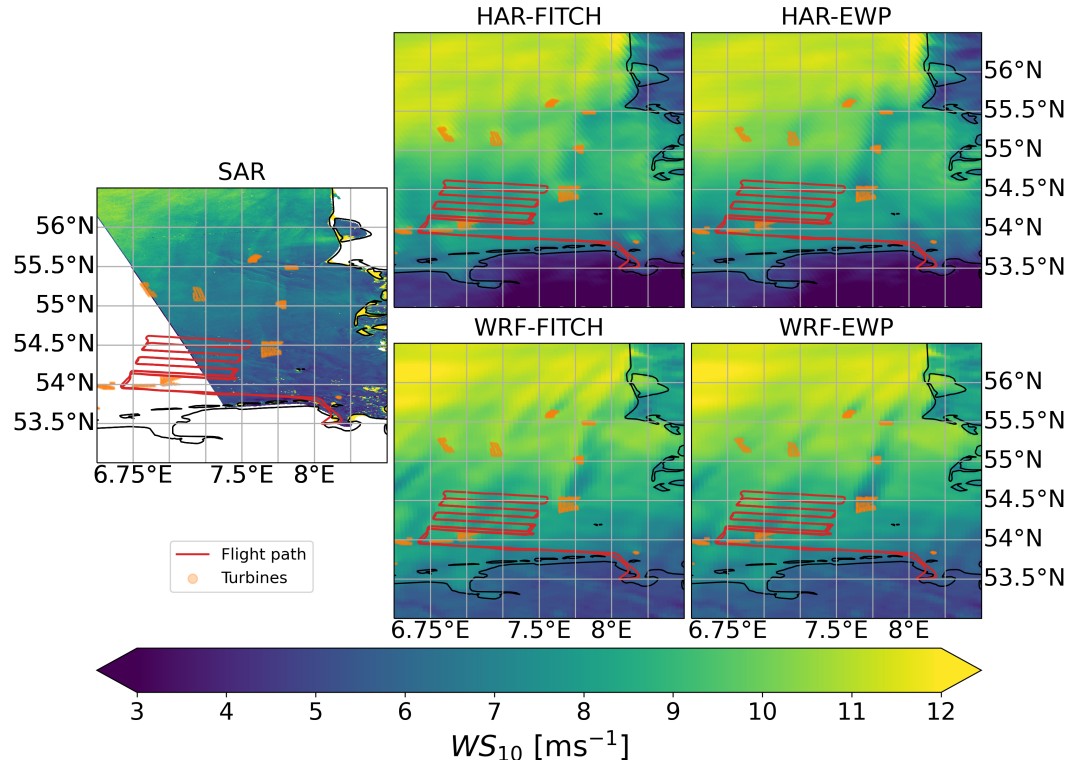

**Figure 9.** Same as Fig. 7, but for 15 October 2017 and for WRF at 17:10, HAR at 17:15, SAR at 17:09.

The meteorological situation on 08 August 2017 is much more dynamic with patches of high and low wind speeds (Fig. 10), but is also classified as slightly stable in Cañadillas et al. (2020). The models can capture this general behaviour, but do not correctly simulate the location of these patches in the 10 m wind with respect to SAR (Fig. 10).

### 3.2.2 Transects in the wake

While the wind speed at 10 m was overestimated by the simulations with respect to SAR for 15 October 2017 (Fig. 9), the wind speed at the transects at 120 m height is quite well matched (Fig. 11). In the sequence of transects, a wind speed reduction in the wake downstream of GodeWind 1+2 and with a smaller amplitude also for Nordsee One is visible in the transects from



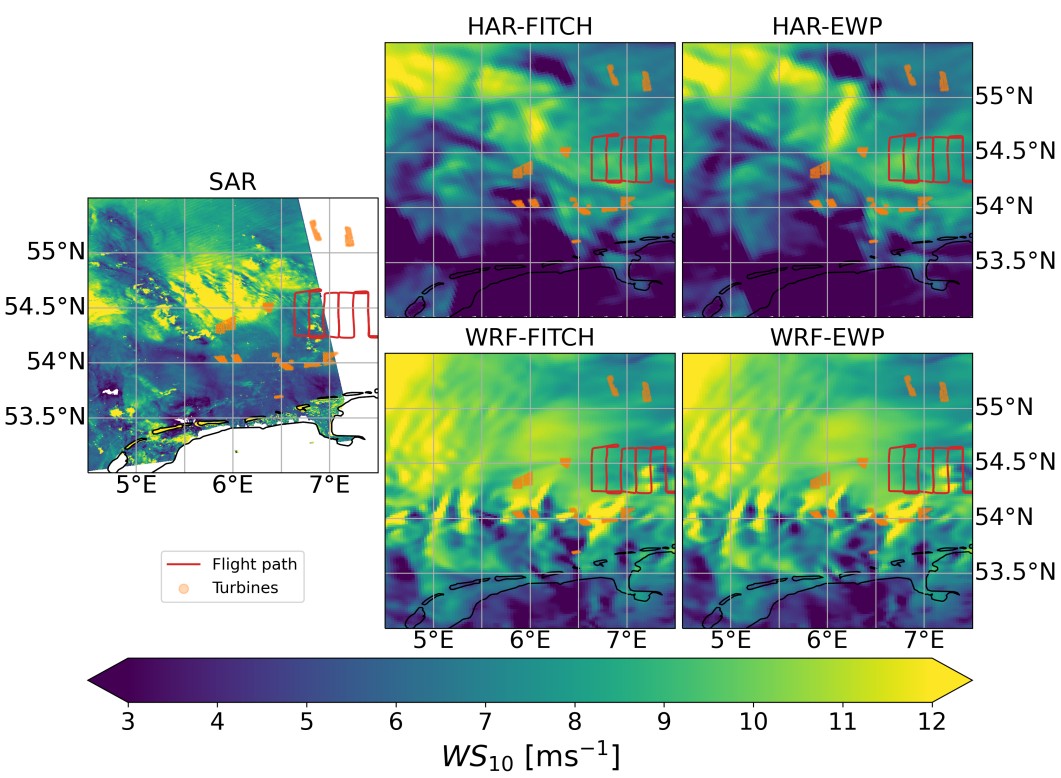

**Figure 10.** Same as Fig. 7, but for 08 August 2017 and for WRF at 17:20, HAR at 17:30, SAR at 17:25.

12:35 onward compared to the transect upstream of the farms at 12:25. The magnitude of the wake deficit gradually decreases with increasing distance from the farm.

TKE is increased downstream of the farm (Fig. 11, third row), especially close to the farm. The TKE increase diminishes

faster with increasing distance from the farm than the $WS$ decrease and is only visible through two peaks around 19 km behind the farm (transect at 13:15). High TKE values are visible especially at the edges of the wake as indicated by the "M"-shape around the wake (e.g. Fig. 11, third row, for 13:05 onward). This increased turbulence is generated by the shear in that region due to the gradient in wind speed inside and outside of the wake. This was also described in Platis et al. (2020).

Both HARMONIE and WRF agree quite well with the observations ahead of the farm and for some flight legs. The wind

direction is slightly more westerly in WRF, which causes a slight displacement of the maximum wind speed deficit at e.g. 30 km (13:25-13:35). Both EWP and FITCH in both models can capture the wake deficit. However, EWP underestimates the magnitude of the wind speed deficit and also strongly underestimates the increased TKE in the wake. HARMONIE-FITCH best captures the magnitude of the TKE increase just downwind of the farm. WRF-FITCH produces slightly smaller magnitudes, although both use a TKE factor of 1. Neither of the models can capture the "M"-Shaped behaviour of the TKE distribution

further downstream the wake. This is expected from the coarse resolution of the mesoscale models.



**Figure 11.** As Fig. 8, but for 15 October 2017 and for around 120 m height. The title in each column indicates the time of the transect and the median downstream distance from the farm.

Fig. 11 only shows the transects up to about 30 km downstream (13:35), since for increasing distance the performance of especially WRF deteriorates due to the increasing offset in the simulated wind direction. As the main objective is to evaluate the performance of the WFP in HARMONIE and WRF and not the background physics, the figure for the later transects is placed in the appendix (Fig. A1). At those transects the trend of increasing wake recovery and decreasing effects on TKE

continues and later closer transects (at 15:05) are again better captured.

For 08 August 2017 the respective figure showing the transects is placed in the appendix (Fig. A2), because the general trends of the results are quite similar to those for 15 October 2017: The magnitudes of the wind speed deficit and TKE enhancement agree better for FITCH than for EWP. Both EWP and FITCH capture the gap between the two farms.





### 3.2.3 Error statistics

We calculate error metrics as described in Sect. 2.6 to quantify the agreement for the different scenarios with observations for the transects highlighted in Fig. 11 and Fig. A2. As for the case above the farm, overall HARMONIE and WRF perform similar (Table 5, Table 6) and again the FITCH agrees best with the observations, indicated by high correlation coefficients and low BIASes for 15 October 2017. For 08 August 2017, FITCH simulations show large BIASes for $WS$ (Table 6) and better performance of EWP. This is due to a systematic underestimation of the wind speed compared to the observations, which is amplified in FITCH and is highlighted by the BIAS. However, looking at the correlation and STDE as a non-systematic error metric, indicates that FITCH also outperforms EWP in this case. This again highlights the challenge of simulating the background meteorology correctly. For TKE FITCH performs best in terms of all error measures.

For Table 5 and Table 6 the mean over all transects is taken. Since this also includes transects with very small TKE-related WFE 20 km and more downstream (Fig. 11, Fig. A2), the difference between EWP and FITCH in the error measures is not that pronounced except for the correlation. This indicates that although EWP greatly underestimates TKE close to the farm, further downstream at hub-height this underestimation is of minor importance due to the diminishing effects for wind farm generated TKE.

**Table 5.** As Table 3, but for case 15 October 2017.

|  |  | WRF | | | HARMONIE | | |
|---|---|---|---|---|---|---|---|
|  |  | FITCH-obs | EWP-obs | NWF-obs | FITCH-obs | EWP-obs | NWF-obs |
| $WS$ | BIAS [ms$^{-1}$] | 0.07 | 0.36 | 0.79 | 0.29 | 0.71 | 1.06 |
|  | CORR | 0.80 | 0.81 | 0.13 | 0.79 | 0.85 | 0.67 |
|  | RMSE [ms$^{-1}$] | 0.68 | 0.67 | 1.27 | 0.86 | 0.92 | 1.29 |
|  | STDE [ms$^{-1}$] | 0.51 | 0.58 | 1.01 | 0.72 | 0.53 | 0.67 |
| TKE | BIAS [m$^2$ s$^{-2}$] | -0.03 | -0.05 | -0.05 | -0.01 | -0.07 | -0.07 |
|  | CORR | 0.49 | -0.31 | -0.27 | 0.46 | -0.24 | -0.15 |
|  | RMSE [m$^2$ s$^{-2}$] | 0.10 | 0.11 | 0.11 | 0.09 | 0.12 | 0.12 |
|  | STDE [m$^2$ s$^{-2}$] | 0.10 | 0.10 | 0.10 | 0.08 | 0.10 | 0.10 |

### 3.2.4 Wind farm effects

The magnitude of the WFEs is derived again by the difference between the simulations with and without WFP. By calculating the mean difference across each transect, the magnitude of the WFE with increasing distance to the farm is derived for the simulations and observations (Fig. 12). For the observations the methods described in Sect. 2.6 are used to generate an artificial NWF-observation. Note that the method by Cañadillas et al. (2020) can only be applied to $WS$ and not to TKE.

Both cases show that TKE (Fig. 12, dotted lines above zero) recovers faster to background levels compared to $WS$ (Fig. 12, solid lines) behind the farm at hub height: after 20-30 km downstream almost no mean TKE effect along the transect is



**Table 6.** As Table 3, but for case 08 August 2017.

| | | WRF | | | HARMONIE | | |
| | | FITCH-obs | EWP-obs | NWF-obs | FITCH-obs | EWP-obs | NWF-obs |
|---|---|---|---|---|---|---|---|
| WS | BIAS [ms$^{-1}$] | -2.27 | -1.66 | -0.79 | -1.81 | -1.16 | -0.58 |
| | CORR | 0.69 | 0.55 | -0.06 | 0.82 | 0.74 | 0.07 |
| | RMSE [ms$^{-1}$] | 2.46 | 2.01 | 1.59 | 1.99 | 1.51 | 1.42 |
| | STDE [ms$^{-1}$] | 1.00 | 1.03 | 1.38 | 0.81 | 0.84 | 1.31 |
| TKE | BIAS [m$^2$ s$^{-2}$] | -0.21 | -0.25 | -0.26 | -0.08 | -0.28 | -0.28 |
| | CORR | 0.79 | 0.27 | -0.31 | 0.74 | -0.07 | -0.47 |
| | RMSE [m$^2$ s$^{-2}$] | 0.28 | 0.33 | 0.34 | 0.18 | 0.36 | 0.36 |
| | STDE [m$^2$ s$^{-2}$] | 0.19 | 0.21 | 0.22 | 0.15 | 0.22 | 0.22 |

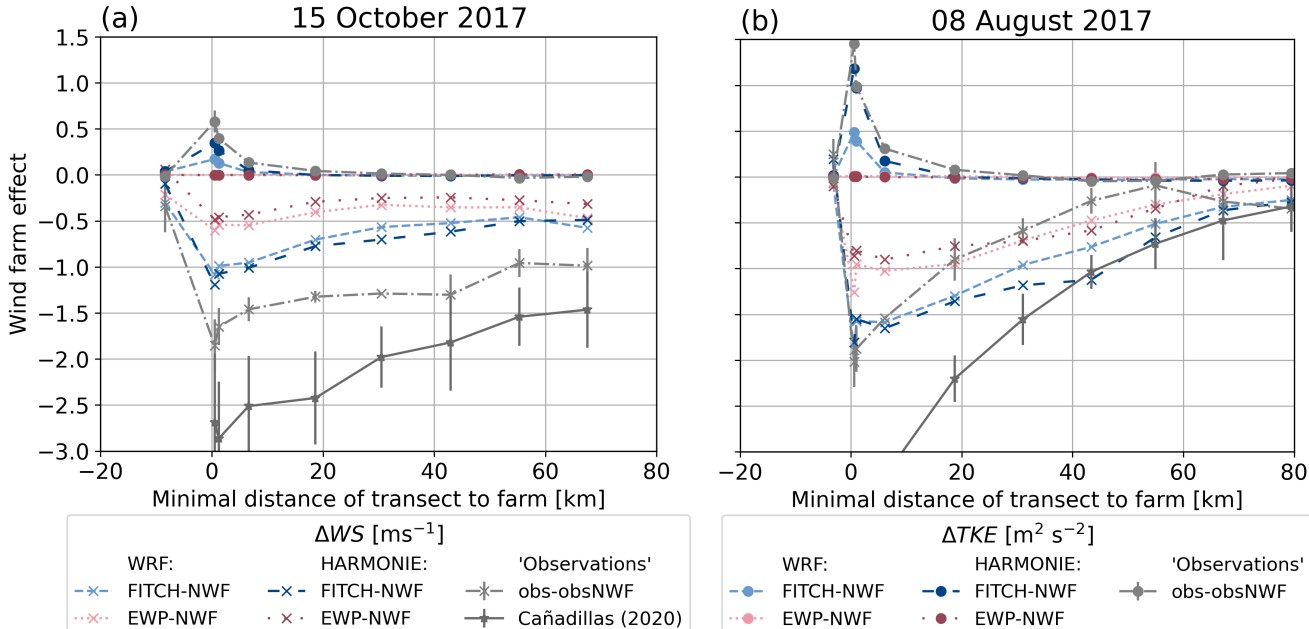

**Figure 12.** Wake effect at around hub height defined as scenario with WFP (FITCH and EWP) minus scenario without wind farm (NWF) in terms of $WS$ reduction (lines with crosses) and TKE increase (lines with circles) for different transects with a certain minimal distance to the farm for WRF and HARMONIE with same color and line style coding as in Fig. 8 for (a) 15 October 2017 and (b) 08 August 2017. Observational wake effects have been derived based on the methods described in Sect. 2.6. The unbiased standard error of the mean is used to draw the error bars around the observational wake effect.

detectable, while wind speed is still reduced compared to the NWF scenario. The wind speed deficit is stronger on 08 August 2017 than on 15 October 2017, but also the recovery is faster, as indicated by the steeper lines. That is also confirmed by the





artificially derived observational WFE and by the exponential function provided by Cañadillas et al. (2020). The WFE based on Cañadillas et al. (2020) are stronger, since they represent maximal WFE, in contrast to the mean WFE in our study.

The WFEs in WRF and HARMONIE are comparable for the two WFPs with slighlty higher values for HARMONIE-FITCH than WRF-FITCH and slightly higher values for WRF-EWP than for HARMONIE-EWP. The similarity confirms the conclusions in Sect. 3.1.4 that the implementation of FITCH and EWP in HARMONIE have been successful.

### 3.2.5 Impact on temperature and humidity

On 08 August 2017 a slight cooling (less than 0.5 K) and humidification (less than 0.5 hPa) was observed in the wake of the farm at hub height a few kilometers downstream of the farm (Fig. 13, compare first with second and third column). This WFE is superimposed by a general cooling and humidifying trend as moving further offshore. On the transects 6 km and further downstream this effect is difficult to detect, since it is super-imposed by the variability in the background conditions. Both HARMONIE and WRF only match the background conditions well for some transects. Therefore, it is difficult to compare the effect quantitatively. However, at least for WRF, out of the three scenarios, EWP best captures this effect, since it shows the largest temperature drop and humidity increase compared to the other scenarios. In HARMONIE the transects upwind of the farm already differ greatly for the different scenarios, thus it is difficult to conclude, whether the WFPs capture this effect. Therefore, only WRF will be used for the more detailed analysis in the following.

The profiles in Fig. 14 show the reason for this WFE. The yellow NWF scenario indicates how profiles evolve with increasing distance from the shore: the inversion moves upward, the air cools and humidifies and wind speed and TKE increase. These changes happen throughout the lower lower atmosphere, e.g. also close to the surface.

Due to the presence of wind farm effects in EWP and FITCH the evolution of these profiles is modified. Compared to NWF, in EWP the inversion height is increased and reaches the upper part of the rotor. This results in lower temperatures and higher vapour pressure at hub height. The weak low-level jet present in NWF is removed through the introduced wind speed deficit from the turbines (Fig. 14d) and TKE levels close to the ground are reduced in EWP compared to NWF. Similarly also temperature, humidity and wind speed are modified close to the ground, indicating that also the surface fluxes have changed. It is difficult to quantify this effect, however, due to the evolving profile with distance from the shore in the NWF case as discussed above.

In contrast to EWP, the strong mixing in FITCH (Fig. 14c) leads to lower inversion height compared to NWF, which is even moved below hub height (Fig. 14a). This causes the cooling that is also visible in FITCH compared to NWF at hub height in the transects in Fig. 13.

## 4 Discussion

The main goal of this study was to derive, how well the implemented WFPs agree in WRF and HARMONIE-AROME to evaluate the implementation of the WFPs in HARMONIE-AROME. Therefore, the set-ups of WRF and HARMONIE-AROME follow best practices for standard use of the two models, respectively (Table 1). Thus, we applied HARMONIE-AROME in



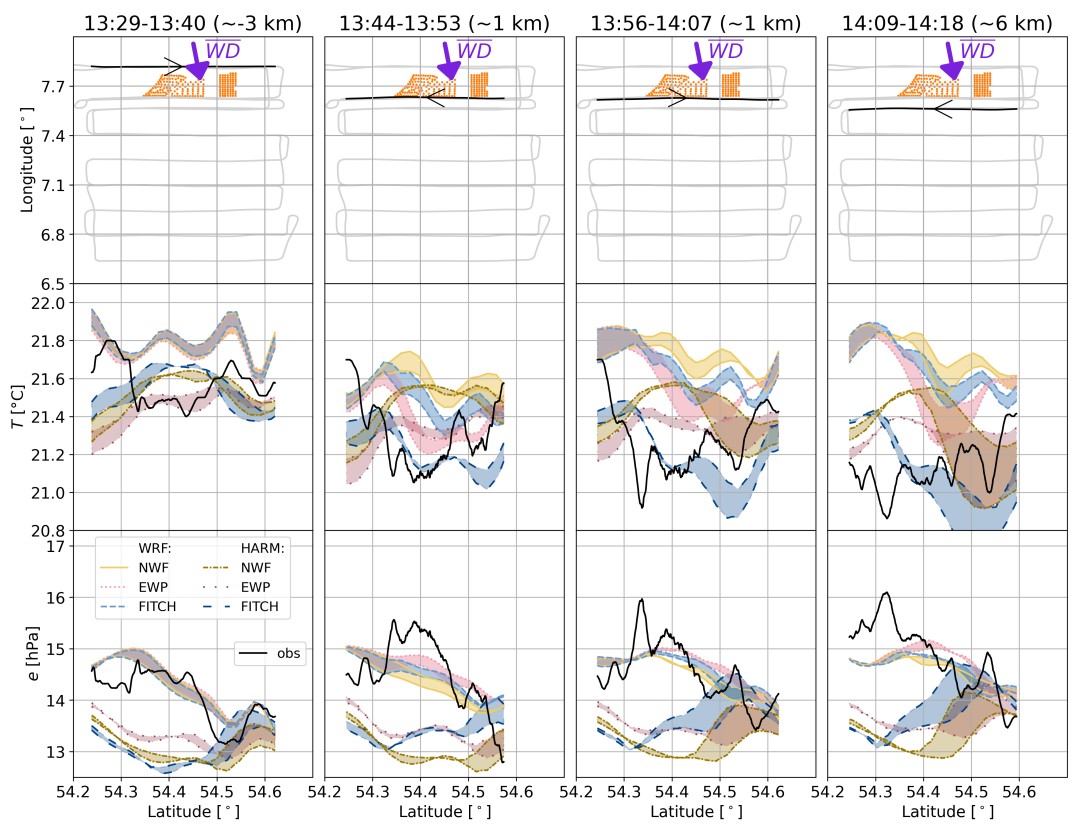

**Figure 13.** As Fig. A2, but for (center row) temperature, (last row) vapour pressure.

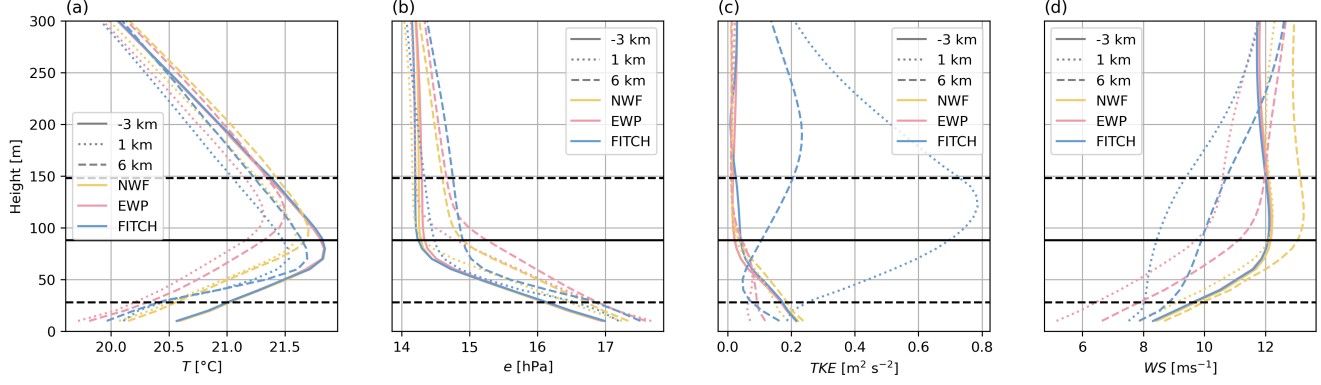

**Figure 14.** Profiles at the transects in Fig. 13 for (a) temperature, (b) vapour pressure, (c) turbulent kinetic energy and (d) wind speed at different distances from the farm (line style) at Latitude 54.4°. All results are for WRF for NWF (yellow), EWP (red) and FITCH (blue).





forecast mode and WRF in hindcast mode, which is a common set-up for wind energy research. Since a control simulation
without wind farms was conducted both for WRF and HARMONIE-AROME, the main goal could be reached even with the
different set-ups. We acknowledge, however, that due to the different resolution and grids, wind turbines have been assigned to
different grid cells (Fig. 5) in WRF and HARMONIE. This influences the direct comparison between WRF and HARMONIE.

The control simulation without wind farms (NWF) for WRF and HARMONIE-AROME was used to derive the magnitude
of the wind farm effects both for $WS$ and TKE as bias compared to a simulation with WFP. To compare these magnitudes
with the research flight measurements, an artificial observation without wind farms was created by simple linear interpolation
between two points outside the farm wake. However, there is some uncertainty in how to define whether a location is outside
the farm wake while still being close enough to not be influenced by other meteorological background effects. This exhibits
uncertainty that was captured by using different artificially produced observations without wind farm effects. However, as
indicated by the NWF simulation, there was considerably variability in the background wind speed within the farm area. Thus,
a linear interpolation can only provide a rough estimate of the magnitude of the wake effect in the observations.

The evaluation of the WFP against real measurements is challenging, since the models ability of simulating the background
meteorological conditions influences the calculations of the WFP: The thrust coefficient depends non-linearly on the wind
speed (Fig. 4) and thus different background conditions will result in different WFEs. Standard operational verification of
HARMONIE-AROME includes mostly observations from automatic weather stations close to the surface. Some evaluation
has also been done for masts (Kangas et al., 2016), but further evaluation of the forecasts from HARMONIE in heights relevant
to wind energy are needed. This evaluation should also include masts undisturbed by wind farms to be able to evaluate the
forecast skill at heights of up to 250 m.

## 5   Conclusions

Wind farm effects are increasingly important to consider in Numerical Weather Prediction (NWP) models. In this study, we
implemented the Explicit Wake Parameterization (EWP, Volker et al., 2015) in the non-hydrostatic NWP model HARMONIE-
AROME. The newly implemented EWP scheme as well as the already implemented (van Stratum et al., 2022) WFP by Fitch
et al. (2012) (FITCH) were evaluated against research flight measurements taken from the project WIPAFF (Bärfuss et al.,
2019) as well as against model simulations with the NWP model WRF, which has been frequently used before to evaluate wind
farm effects (Fischereit et al., 2022).

The results show that the implementation of EWP and FITCH are successful and that, of the two WFPs, FITCH agrees in
general better with measurements, especially for TKE. Most note-worthy is the underestimation of TKE by EWP close to the
farm. The high values of TKE decrease fast with increasing distance of the farm, leaving an "M"-shape pattern with high TKE
values close to the edge of the wake. Because of this fast decrease, the underestimation of TKE by EWP is of minor importance
further downstream. Nevertheless, according to this study EWP shows possibilities of improvement that will be addressed in
future work.



As the next step, forecasts with all wind turbines, both onshore and offshore, within the Northern Europe DMI domain will be performed for longer periods. This allows to estimate the full impact of currently installed wind turbines on weather and weather forecasting. The established wind turbine data base as well as the results will be presented in part 2 of this series (Fischereit et al., 2023a).

*Code and data availability.* The ALADIN and HIRLAM consortia cooperate on the development of a shared system of model codes. The HARMONIE-AROME model configuration forms part of this shared ALADIN-HIRLAM system. According to the ALADIN-HIRLAM collaboration agreement, all members of the ALADIN and HIRLAM consortia are allowed to license the shared ALADIN-HIRLAM codes to non-anonymous requests within their home country for non-commercial research. Access to the full HARMONIE-AROME codes can be obtained by contacting one of the member institutes of the HIRLAM consortium (see Hiram.org). The code-changes to enable wind farms in
HARMONIE-AROME are made available in the supplement material.

The WRF model is available from https://github.com/wrf-model/WRF. The modifications to WRF for EWP are made avilable in the supplement material. Updates to EWP will be made available in the future at https://gitlab.windenergy.dtu.dk/WRF/EWP.

The flight measurements are available from Bärfuss et al. (2019). The SAR data is available from https://science.globalwindatlas.info/ (Badger et al., 2022). ERA5 data is available from Hersbach et al. (2018) and OSTIA data is available from http://my.cmems-du.eu/
motu-web/Motu.

The wind farm input data as well as the namelists for WRF are permanently archived at Fischereit et al. (2023b) along with the scripts to reproduce the tables and figures in this manuscript.

*Author contributions.* JF implemented the EWP parameteriztion with the help of NT and HV. JF designed the experiments, performed the simulations and analysed the results. XG post-processed the flight measurements. GG and EK acquired funding for this research. JF prepared
the manuscript with contributions from XG and HV. All co-authors discussed the analysis, reviewed and edited the manuscript.

*Competing interests.* The authors declare no competing interests.

*Acknowledgements.* Part of the funding is by the Danish state through the National Centre for Climate Research (NCKF). We are grateful for the open access measurements in Bärfuss et al. (2019) from the project WIPAFF (wind park far field). We would like to thank the
members of the NWP group at DMI for their helpful advice on running HARMONIE. Data processing and visualization for this study was in part conducted using the python programming language and involved use of the following software packages: NumPy (van der Walt





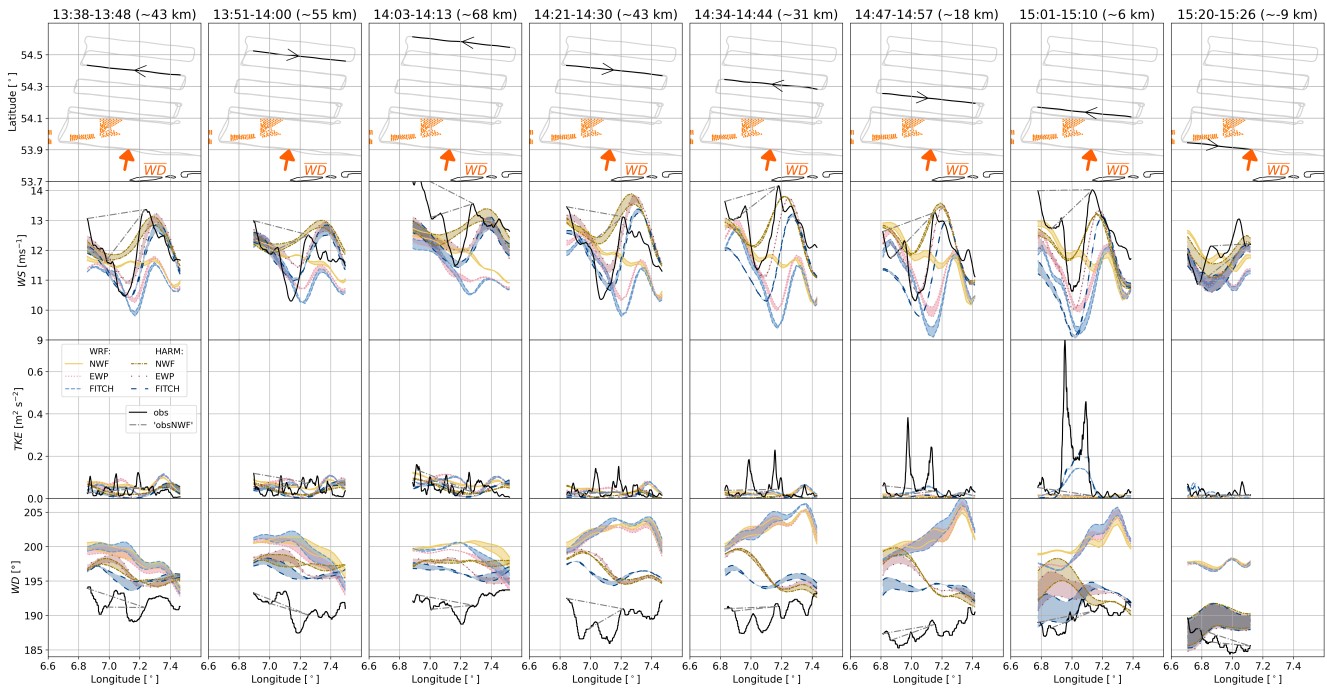

**Figure A1.** As Fig. 11, but for later flight transects.

et al., 2011), pandas (McKinney, 2010), xarray (Hoyer and Hamman, 2017), Matplotlib (Hunter, 2007) and Cartopy (Met Office, 2015). The authors are grateful for the tools provided by the open-source community. The color scheme for some figures are taken from Paul Tol's notes (https://personal.sron.nl/~pault/, last accessed: 31.05.2023)







**Figure A2.** As Fig. 8, but for 08 August 2017 and for around 90 m height.

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
