# Peer review of "Modelling wind farm effects in HARMONIE-AROME (cycle 43.2.2) – part 1: Implementation and evaluation"

_Geoscientific Model Development, 2023_

## Author Response (AR1)

**Response to the reviewers**

We would like to thank the reviewers for their constructive comments. We responded in detail to them below in blue.

**Reviewer 1**

*The article describes an evaluation of an explicit wake parameterization (EWP) in the HARMONIE-AROME mesoscale model. The authors remark that several European weather services already use the HARMONIE-AROME model. Therefore, it is vital to implement wind farm parameterization for this model to allow researchers to work with a familiar code when accounting for wind farm effects in a forecast.*

*Since the technical description of wind farm parameterizations in HARMONIE-AROME was already published, this article focuses more on comparing and presenting the results. The article studying one-day simulation cases also acts as a link between a technical description of wind farm parameterizations in HARMONIE-AROME and an evaluation of long-term simulation effects expected in the second part.*

*Considering the number of graphs presenting the results, the authors did a good job optimizing the layout and description of the figures. Except for a few concerns listed further, the captions are presented in a way that makes easy to understand the plots without looking for details in the main text.*

*The simulations setups and choices made are meticuosly documented. The concepts unique to the study are appropriately introduced first. For example, a double line plot showing the implementation of no wind farm state for the observations is introduced separately in Fig. 6; this figure is further referred to in the caption of composite figures, so a reader can quickly check what the straight dashed lines mean. The caption of Fig. 8, one of the most complex figures in the article, also helps to recall the explanation immediately without searching for it in the main text.*

*The evaluation puts the EWP and FITCH parameterization in HARMONIE-AROME into a broader context by comparing their performance to the same parameterizations in the WRF model (the most widely used mesoscale model) and observations. This way of comparing allows other researchers to make an informed choice when selecting a mesoscale model to work with wind farm effects.*

*My suggestions for the revisions would be, therefore, minor leaning to technical.*

*Minor revisions*

*1. **Page 14-15, lines 240-244** and Conclusion mention a possible effect of different grids on the direct comparison of WRF and HARMONIE highlighted by wind farm placement in **Fig. 5**. Compared to other complications, such as a choice of TKE correction factor or artificial no-farm observations, this problem is addressed only briefly as a possible source of the differences in predicted wake effects and leaves a question if there are other differences between models that could have been caused by it. Is it possible to quantify this effect?*

This is indeed an important point. Unfortunately, it is not possible to align the grids for HARMONIE and WRF, since WRF uses a nested domain setup, while HARMONIE uses a uniform resolution throughout the entire study area. As we discussed, the idea was to follow best practices when setting up the model domain, i.e. for WRF following the best practices in the wind energy community and for HARMONIE following the operational NWP setup at DMI. The differences due to the grids cannot be easily quantified. However, for our analysis where we compare wind farm parameterizations results for WRF and HARMONIE directly, i.e. Table 4 and Figure 12, we use a no-wind-farm simulation to normalize the wind farm effect. This should remove the issue of the different grids for this part of the analysis.

We have added the following text in the discussion: "In addition, the different grids in WRF and HARMONIE could also affect the modelling of the background meteorology, which has implications for comparison against real measurements as discussed below. It is difficult to isolate and quantify the effects of the different grids directly, since also the physics schemes as well as the initial and boundary data differs between WRF and HARMONIE (Table 1). More idealized test cases and setups with WRF and HARMONIE could be used in the future to compare WRF and HARMONIE directly and isolate some effects, such as differences in the grids and in the physics schemes."

*I like the colour coding for tables – it makes them easy to read and interpret. Nevertheless, the colour-coding may not be quite in line with the article text or cause an erroneous interpretation. It is need to be checked whether the tables communicate the main idea properly, so I'll list my concerns.*

*2. The colour tones in tables may be misleading in some cases. Particularly, the TKE STDE row of **Page 19, Table 5**: the values of 0.10 are coloured in different shades of red, while 0.08 is white. Considering the rounding to two digits, one would expect all 0.10 to be coloured the same. Otherwise, it appears as if the difference in a third digit is important but cannot be seen due to the round-up. Similar behaviour may be observed in the TKE BIAS, RMSE rows of **Page 19, Table 5** and, to a lesser extent, in the TKE RMSE, STDE rows of **Page 20, Table 6**.*

We fully agree with the reviewer, this behavior was not intended from our side. We now changed the color coding: now it only relies on the significant digits.

*3. The colour tones also raise a question of whether the difference in the TKE STDE between 0.08 and 0.10 in **Table 5** is as important as in the TKE STDE between 0.29 and 0.55 in **Table 3**, considering the similar shading. It is also noted in the article text that the models' agreement with observations in the wake for 15 October 2017 (**Table 5**) is noticeably better than above the wind farm for 14 October 2017 (**Table 3**). However, similar tones imply equally strong differences for the TKE BIAS, RMSE and STDE in **Table 3** and in **Table 5**.*

We agree that for comparing the different dates, it is useful to use a consistent color-coding across all days. We have adjusted the tables accordingly (see Figure 1 in this response) and we also added to the table description of Table 3: "across all three case studies".

**Table 5.** As Table 3, but for case 15 October 2017.

| | | WRF | | | HARMONIE | | |
| | | FITCH-obs | EWP-obs | NWF-obs | FITCH-obs | EWP-obs | NWF-obs |
|---|---|---|---|---|---|---|---|
| $WS$ | BIAS [ms$^{-1}$] | 0.07 | 0.36 | 0.79 | 0.29 | 0.71 | 1.06 |
| | CORR | 0.80 | 0.81 | 0.13 | 0.79 | 0.85 | 0.67 |
| | RMSE [ms$^{-1}$] | 0.68 | 0.67 | 1.27 | 0.86 | 0.92 | 1.29 |
| | STDE [ms$^{-1}$] | 0.51 | 0.58 | 1.01 | 0.72 | 0.53 | 0.67 |
| TKE | BIAS [m$^2$ s$^{-2}$] | -0.03 | -0.05 | -0.05 | -0.01 | -0.07 | -0.07 |
| | CORR | 0.49 | -0.31 | -0.27 | 0.46 | -0.24 | -0.15 |
| | RMSE [m$^2$ s$^{-2}$] | 0.10 | 0.11 | 0.11 | 0.09 | 0.12 | 0.12 |
| | STDE [m$^2$ s$^{-2}$] | 0.10 | 0.10 | 0.10 | 0.08 | 0.10 | 0.10 |

**Table 6.** As Table 3, but for case 08 August 2017.

| | | WRF | | | HARMONIE | | |
| | | FITCH-obs | EWP-obs | NWF-obs | FITCH-obs | EWP-obs | NWF-obs |
|---|---|---|---|---|---|---|---|
| $WS$ | BIAS [ms$^{-1}$] | -2.27 | -1.66 | -0.79 | -1.81 | -1.16 | -0.58 |
| | CORR | 0.69 | 0.55 | -0.06 | 0.82 | 0.74 | 0.07 |
| | RMSE [ms$^{-1}$] | 2.46 | 2.01 | 1.59 | 1.99 | 1.51 | 1.42 |
| | STDE [ms$^{-1}$] | 1.00 | 1.03 | 1.38 | 0.81 | 0.84 | 1.31 |
| TKE | BIAS [m$^2$ s$^{-2}$] | -0.21 | -0.25 | -0.26 | -0.08 | -0.28 | -0.28 |
| | CORR | 0.79 | 0.27 | -0.31 | 0.74 | -0.07 | -0.47 |
| | RMSE [m$^2$ s$^{-2}$] | 0.28 | 0.33 | 0.34 | 0.18 | 0.36 | 0.36 |
| | STDE [m$^2$ s$^{-2}$] | 0.19 | 0.21 | 0.22 | 0.15 | 0.22 | 0.22 |

*Figure 1: Updated tables with same color coding across dates and relying on 2 significant digits only.*

*4. Another confusing usage of colour is the BIAS row in all colour-coded tables. While other rows set the colour map to highlight the best case regardless of its value, the BIAS row shows the absolute deviation from zero. I may be missing some common convention here. If this is the case, I would be glad for a clarification.*

We are not aware of any convention. However, following our approach for the other metrics, we also highlighted the best case for the BIAS with the whitest color. As we now use color-coding across all days, this is now better visible: the BIASes are negative in Table 6, but positive in Table 3 and 5, but the same color shading is applied for instance for the WS BIAS for NWF-obs in Table 5 (0.79 m/s) and Table 6 (-0.79 m/s) (see Figure 1 in this response).

**Technical corrections**

*I will try to focus on the typos which are harder to track down in the proofreading.*

*1. The commas are occasionally omitted for introductory phrases.*

*E.g.,* **Page 2, line 49-50** *does not have a comma between 'atmosphere' and 'EWP':*

*To parameterize the effect of wind farms on the atmosphere EWP imposes an elevated momentum sink or drag force on a control volume...*

*However,* **Page 3, lines 54-55** *have a comma between 'effect' and 'EWP' in a similarly constructed sentence:*

*To account for this effect, EWP builds upon classical wind turbine wake theory, by assuming an exponential expansion based on an effective length scale...*

*In general, the punctuation for introductory phrases allows some freedom. A comma can usually be omitted for short introductory phrases and is preferable for long phrases, but its usage is the opposite here. I suggest double-checking the text and comma rules to ensure that commas are used consistently.*

Thank you for pointing this out. We went through the text and adjusted the commas as suggested.

*2. **Page 3, line 78**: an uncommon placement of 'not'*

*For the implementation of Eq. 4 not the true wind direction WD is used...*

*Probably, it was supposed to be*

*For the implementation of Eq. 4, the true wind direction WD is not used…*

Thank you for this suggestion. The sentence now reads "For the implementation of Eq. 4, $WD$ is not the true wind direction, since u and v are grid-following in HARMONIE, and are thus not necessarily aligned with the cardinal directions north-south and east-west."

*3. **Page 6, Table 1**: FITCH is not mentioned for WRF in the title row, but WRF is also used with FITCH implemented, isn't it?*

Indeed, we also use WRF with FITCH as shown in the last row. However, since WRF 4.2.2 includes FITCH by default, we do not mention it in the "Version" row of the table.

*4. **Page 8, Fig. 3**: this could be my monitor calibration, but bright and dark colours are harder to distinguish for 08 August 2017 lines than for other tracks-transect pairs. The flight track for 15 October 2017 may need to be slightly darker to avoid blending with the background.*

Thank you for these suggestions. Unfortunately, it was not possible to adjust the colours slightly, since this caused issues in color-blind mode. Instead, we used a new set of colors entirely. Due to that, we also adjusted the main wind speed arrow in Figure 6, Figure 8, Figure 11, Figure 13, Figure A1 and Figure A2. The new color scheme is shown in Figure 2 in this document.

[Figure]

*Figure 2: Figure 3 with new color coding for better visibility.*

*5. **Page 8, line 150**: the dates are typed as 14.08.2017 and 15.08.2017. Are they supposed to be 14 October 2017 and 15 October 2017, the cases regarded in the article?*

Thanks for catching this. We adjusted it as suggested.

*6. **Page 9, Fig. 5**: a typo in the caption '...the winf farms…'*

Corrected.

*7. **Page 13, line 218**: an uncommon placement of 'also', similar to the one mentioned in (3)*

*As a consequence also the exact location of the wind speed deficit is not as well captured...*

*Please, double-check the text for similar errors.*

We have changed the sentence as follows "A consequence is that the exact location of the wind speed deficit is not as well captured in HARMONIE as it is in WRF".

We also went through the entire text to find other errors. In addition, we also corrected some other English phrases throughout the text.

*8. **Pages 14-15**: the captions of **Fig. 8** and **Tables 3-4** do not mention the date and height of the case shown – it requires remembering the case's specifics to compare them to similar figures/tables.*

We added the information to the captions.

*9. **Page 21, lines 323-324**: duplicated 'lower'*

*These changes happen throughout the lower lower atmosphere, e.g. also close to the surface.*

Thanks. We removed one "lower". We also removed a duplicated "that" in line 232 of the original manuscript.

*10. **Page 21, Section 4** and further: As noted in the Introduction, HARMONIE-AROME is shortened to just HARMONIE. However, **Sections 4-5** use the full name again and occasionally switch to the shortened name. Is there any specific reason behind this?*

We changed HARMONIE-AROME in sections 4 and 5 to HARMONIE to make it consistent.

*11. Cross-referencing figures in the captions shortens the description but also forces scrolling back and forth or searching in the text to find out which case is shown in the figure. Considering the amount of information and the size of figures detaching them from the first mention in the text, it is possible to get lost. I expected to see captions similar to **Fig. 11** for all figures using the same layout as **Fig. 8**: reference to Fig. 8 for the full description, correction for the date and height, and additional comment if needed.*

*However, several captions break the pattern:*

*- **Figure 13** in the main body references Fig. A2 in the Appendix (= needs scrolling down to confirm which case is this or searching the text), which in turn references Fig. 8 but adds date and height. It would be easier to read if Fig. 13 also referenced Fig. 8 with the added date and height.*

*- Similarly for **Fig. A1**, its caption does not mention the new date and height and requires to scroll up to see the case.*

Thank you for the detailed revision. We have changed the captions as suggested.

*12. The wind farms are referred to by name several times. Although it is not very ambiguous from the text + figures which farm is located where, it would be nice to see an overview figure of the wind farm placement.*

We added the names of relevant farms to Figure 3 (see Figure 2 in this document). We also added Meerwind Süd/Ost in Table 2  for the case on 08 August 2017, since the southern part of the flight track is also influenced by this farm. We also modified section 2.5 accordingly.

*13. **Page 24, line 379**: a typo in the URL Hiram.org – it should be Hirlam.org, isn't it? Also, it is not possible to access this site for an external user. The page only asks for a login and password, unlike the EWP GitLab or CMEMS data pages, which are also account-locked but provide information on acquiring access. If there exists a different page with the relevant contact information for HIRLAM consortia, consider adding it instead.*

*hirlam.org is also mentioned twice in the main text, but it is accompanied by a citation, so this URL does not require a change in the main text.*

We have replaced the url with [https://hirlam.github.io/HarmonieSystemDocumentation](https://hirlam.github.io/HarmonieSystemDocumentation), which should provide relevant information. We felt like it would be more useful to have an accessible link throughout the manuscript, thus we also decided to replace hirlam.org in the main text.

*14. I do not see the Appendix section's title even though two Appendix figures are present. It could be a glitch in the template.*

Thanks for pointing this out. We fixed this.

**Reviewer 2**

*The authors presented a comparison between the Fitch wind farm parameterization and the EWP within two models: WRF and HARMONIE. The results of both models are compared to some airborne measurements and to SAR. A control experiment for both parameterizations with no wind farms is used as a reference case. The authors provided their setup and post-processing files publicly, which is a credited effort and can be helpful to other researchers to try regenerate the same results.*

*In general, I was not convinced that this manuscript can be a standalone publication, and I recommend a major revision. My main comments are listed below, and hopefully they be of help to the authors.*

*1. The objective of the paper is to compare the performance of Fitch and EWP in the model HARMONIE. The major concern here is:*

*how is this different from previously published studies that compared Fitch and EWP in WRF?*

*For example, Pryor et al (2020, https://doi.org/10.1175/JAMC-D-19-0235.1) did a similar comparison as well as some of the authors of this manuscript (Larsén and Fischereit, 2021, https://doi.org/10.5194/gmd-14-3141-2021).*

*Using a different model (WRF vs HARMONIE) is irrelevant in this context. Both Fitch and EWP depend on the incoming flow (e.g., wind speed, wind direction, turbulence, etc) whether this flow is resolved by WRF or by HARMONIE. It is true that both models (WRF and HARMONIE) will resolve different flow fields and will transport the wake of the wind farms differently, but this is a comparison between WRF and HARMONIE, which is not the purpose of this study. As far as Fitch and EWP are concerned, the results and the conclusions presented here are not new to what is already known in the literature using WRF simulations.*

*It is stated that this manuscript is a part of a series of manuscripts using HARMONIE to model all the wind turbines in Europe. I agree with the authors that the first order of business is to make sure that Fitch and EWP are correctly implemented by comparing them to field measurements and to other models (in this case WRF). However, I suggest that this manuscript be summarized and included as an implementation-check-up section in upcoming manuscripts.*

Thank you for raising your concerns. The main objective of this paper is not only to compare FITCH and EWP, but, as stated in line 36 of the original manuscript, the manuscript aims to describe "the implementation of EWP in HARMONIE as well as the comparison against WRF results and flight measurements for three case studies". Thus, the purpose of the manuscript is actually three-fold:

1. Ensuring that EWP (and FITCH) are correctly implemented in HARMONIE

2. Evaluate how wind farm effects are manifested and transported in HARMONIE and verify that by comparing to WRF

3. Check how well the wind farm effects agree with a unique data set of observations in the wind farm wake.

Thus, this manuscript goes significantly beyond just the implementation part of EWP in HARMONIE. However, we acknowledge that these three-fold objectives wer not clear enough from the previous manuscript version. Therefore, we now made the objectives of this first part of the paper series more clear by adding the following in the introduction:

"Part 1 in the present manuscript has three objectives: (1) Ensuring that EWP and FITCH are correctly implemented in HARMONIE, (2) evaluate how wind farm effects are manifested and transported in HARMONIE and verify that by comparing to WRF and (3) check how well the wind farm effects in both HARMONIE and WRF agree with flight measurements in the wake and above wind farms. To archive those objectives the manuscript is structured as follows: [...]"

We agree with the reviewer that the main conclusions regarding the difference in wind speed deficit and turbulent kinetic energy behaviour with respect to EWP and FITCH are already known through studies with WRF. We acknowledge that existing literature on this topic has not been sufficiently cited, which we have addressed in the updated version (see also the response to point 2). However, what makes this study unique, is the use of measurements in the wake of wind farms that have not been explored to so far when comparing EWP and FITCH. Previous literature, including the mentioned manuscript of Pryor et al (2020, https://doi.org/10.1175/JAMC-D-19-0235.1) discussed the differences of the two schemes, without evaluating the performance against actual measurements. The previous study by some of the authors of this study (Larsén and Fischereit, 2021, https://doi.org/10.5194/gmd-14-3141-2021) did compare against measurements, but only for a case study above a wind farm and not in the wake.

Finally, concerning the comparison between HARMONIE and WRF, we agree that this comparison could be and should be done independently as well, but wind farm wakes provide a good tool to understand the differences in HARMONIE and WRF regarding the flow field and TKE advection. The availability of measurements within the wind farm wake, make it a good opportunity to compare HARMONIE and WRF with respect to these aspects. In addition it is an important finding for the forecasting community, especially for those centers that use HARMONIE, that the behavior of EWP and FITCH in HARMONIE agrees with the behavior in WRF.

Overall, we think that this content is sufficient for a standalone manuscript as it provides standalone conclusions and will greatly lengthen the second part of the paper series, if integrated.

*2. As a follow up to the last point, the introduction section misses any previously published differences between Fitch and EWP. It will be helpful for the reader that a study focusing on a comparison between Fitch and EWP discusses what the literature has to say about this issue.*

We agree with the reviewer that this aspect is missing in the introduction. We have now added another paragraph as follows:

"Previous studies have explored the differences between wind farm effects predicted by FITCH and EWP in WRF. Pryor et al. (2020) noted in their nine month long study of the U.S. Midwest that

capacity factors were lower for simulations with FITCH than with EWP. They also found that wind speed deficits and TKE enhancements extended over a larger area for FITCH than for EWP. Similar conclusions were drawn by Shepherd et al. (2020), who performed a yearlong simulation for Iowa. They also noted that the differences lead to differences in impacts on near-surface climate variables. Fischereit et al. (2022b) compared high-resolution RANS simulations with WRF+WFP simulations and noted that EWP underestimated the wake wind speed deficits between farms, while FITCH performed reasonably well. Larsén and Fischereit (2021) found that above a wind farm both EWP and FITCH can capture the wind speed deficit fairly well compared to measurements. However, EWP significantly underestimated TKE above the farm. This study builds upon and extends the previous studies on comparing EWP and FITCH by comparing with actual measurements within the wake."

And in the conclusion section we added:

"Most note-worthy is the underestimation of TKE by EWP close to the farm, which has also been reported previously (Larsén and Fischereit, 2021)."

*3. Can you please elaborate more on the annotations of the variables in equations 2—4? For instance, what does a subscript (rh) mean. What is an overbar? Even though these equations are explained in more detail in the literature (e.g. Volker et al 2015), for closure it is best to explain the meaning of the used variables.*

Thanks for pointing out that some variables were not explained properly. We have now added the description in line 55 – 60 of the new manuscript:

"To account for this effect, EWP builds upon classical wind turbine wake theory, by assuming an exponential expansion based on an effective length scale $\sigma_e$ to define the drag force at grid cell x, y, z [...] Here, Nt is the number of turbines per grid cell, $r_h$ is the rotor radius = 0.5d with d being the rotor diameter, $\overline{u_{rhxy}}$ denotes the horizontal wind speed $\overline{u_r} = \left(\overline{u_1}^2 + \overline{u_2}^2\right)^{0.5}$ with 1 and 2 denoting the wind components in the two horizontal directions at hub height h averaged over one grid cell and finite time increment, as indicated by the overbar, $z_z$ is the height of the model level z and the thrust [...]"

*4. In line 114: "The later initialization causes the simulations follow more closely the boundary conditions":*

*Can you please elaborate more what this means? A later initialization means a shorter spin-up period. If this means that after 12 hours of initialization, the model starts to deviate significantly from field measurements, does this raise a concern about the model setup? Can you please explain in case I understood this incorrectly?*

This was indeed misleadingly formulated. We have now reformulated the sentence as "The sensitivity test show that for stable cases the later initialization helps to better capture transient meteorological conditions by properly introducing the initial conditions to the simulation."

*5. Line 167: "Previous studies found that simulation results are not very sensitive to variation in fr between 1.5 and 1.7":*

*Which "simulation results" are not sensitive? Wind speed, power production, TKE, ..? This study (https://doi.org/10.1175/MWR-D-23-0006.1) showed different conclusions. Can you please provide a proof that the range 1.5 –1.7 does not cause a significant difference at least within the context of the current comparison?*

Thank you for making us aware of this study by Ali et al. (2023), which was not yet published, when we submitted the manuscript in March. In their study they use three different values for $f_r$: 1.36, 1.7 and 2.04. This is a much wider range than the range of 1.5 – 1.7 that we mention in our study. As we highlighted in the original manuscript, the small sensitivity of the simulation results for wind speed within the range of 1.5 – 1.7 was noted in "previous studies (e.g. Volker et al., 2015; Larsén and Fischereit, 2021)" (line 169).

We agree that it could be interesting to test values outside this range. However, this goes beyond the main objective of this study, which was to implement EWP with the best practice that existed so far. The value of 1.7 was used in many other published studies as well, (Pryor et al. 2020 https://doi.org/10.1175/JAMC-D-19-0235.1, Volker et al. 2017, https://doi.org/10.1088/1748-9326/aa5d86).

To make sure that the previously noted low sensitivity within the range 1.5 – 1.7, is also applicable to our study, we made one test simulation with WRF for 15 October. We reproduced Figure 11 from our manuscript, but with reduced complexity, i.e. without the lines for HARMONIE, in Figure 3 of this document. EWP simulations with a coefficient of fr=1.5 (EWP15) are shown in green. Compared to the simulations with fr=1.7 slight differences can be seen in wind speed, especially for the transect close to the farm (~ 1 km), while with increasing distance from the farm this difference reduces. The maximal bias for the transects for EWP1.7 and EWP1.5 in Figure 1 is 0.07 m/s. Thus, the differences are much smaller than the difference compared to simulations with the FITCH parameterization or the no-wind-farm-scenario.

We modified the original sentences as follows: "For EWP we set the tuneable initial wake expansion coefficient, as introduced in Eq. 3, to fr =1.7, as recommended in Volker et al. (2015) and used in other studies (Volker et al., 2017; Pryor et al., 2020). Previous studies (e.g. Volker et al., 2015; Larsén and Fischereit, 2021) found that simulated wind speed deficits in the wake are not very sensitive to variation in fr between 1.5 and 1.7. To confirm the low sensitivity within this range, we reproduced the case 15 October 2017 with fr =1.5 and found only minor biases < $0.1 ms^{-1}$ close to the farm (not shown). Values for fr outside the range of 1.5 and 1.7 were tested in Ali et al. (2023). They noted larger impacts for the tested values of 1.36 and 2.04 on wake length and wind speed deficit. However, a more detailed investigation of different fr-values is beyond the scope of the current study, which focuses on existing best practices.".

[Figure]

*Figure 3: As Figure 11 from the original manuscript, but without the simulations by HARMONIE and with EWP15, EWP with fr set to 1.5, added.*

6. *Line 203: "while WRF slightly overestimates the wind"*

*Can you please elaborate why? This does not seem to be the case in Larsén and Fischereit (2021, Fig. 1) for the same day (14 October 2017) and for only a 10-min time difference (17:10 there against 17:20 here). Can you please explain this discrepancy?*

There are multiple reasons for the difference. Larsén and Fischereit (2021) already documented that it is a quite challenging case to model, due to the transient conditions and the low-level jet developing. While Larsén and Fischereit (2021) could optimize their setup for one particular case, in this study the setup had to provide reasonable results across three very different cases. The most important differences between our setup and the setup in Larsén and Fischereit (2021) are:

- The nested domain setup differs: Compare Fig. 4 in Larsén and Fischereit (2021) and Fig. 1 in this study. This is best visible for the inner-most domain, which extends further east, since on the case study on 08 August winds were from east. To ensure that these winds can properly develop, the inner-most domain was extended further east.

- The WRF version differs. Larsén and Fischereit (2021) used version 3.7.1, while we use version 4.2.2. Differences during the updates from 3.7.1 to 4.2.2 can be seen from https://github.com/wrf-model/WRF/releases.

- We apply spectral nudging above the boundary layer (Table 1), while Larsén and Fischereit (2021) did not use nudging. Nudging was used, since through initial tests that we performed for the current setup, nudging was found to be beneficial with respect to comparison with measurements when comparing all three cases.

7. *Figures 8 and 11 are difficult to read, especially the wind direction row. Can you please make it clearer by making the sub-figures bigger? Maybe split this into multiple figures?*

Thank you for this suggestion. We split the figure into two. We also increased the font sizes in Figure 6 and Figure A2. In Figure 13 and A1, we increased the font size and moved the legend outside the figure. The new figure 8 is shown in Figure 4 of this document as an example.

[Figure]

*Figure 4: Updated Figure 8 according to the reviewer comments*

*8. Line 217: "The wind direction, WD, is slightly off in HARMONIE, especially in the earlier transects up to 15:50"*

*Any suggestions or explanations why this happened?*

As discussed in our response to 6 already, the meteorological conditions during the case study on 14 October 2017 are quite challenging for numerical weather prediction. Weather prediction can never be completely accurate, since the initial state of the complete atmosphere is not known and models rely on approximations. We still think that the simulations capture the meteorological situation well enough to compare the results to the measurements. At the later transects, the agreement for the wind direction is even better in HARMONIE than in WRF, although it still cannot capture the full wind direction variability visible at the transect.

*9. In Fig. 9, wakes are much shorter in HARMONIE (top two sub-figures) compared to WRF (bottom two). Also, the direction of the wake in HARMONIE is a bit off compared to WRF and to the wind direction indicated in Fig. 8. Can you please explain why?*

Fig. 9 refers to the case study on 15 October 2017, while Fig. 8 refers to the case study on 14 October 2017. The equivalent figure to Fig. 8 for 15 October 2017 is Fig. 11. Fig. 11 shows the agreement of the simulations with the flight measurements. From that figure, it is clear that the wind direction at 120 m height from the measurements is actually captured better in HARMONIE than in WRF, which predicts winds more from the west. This is also reflected in the SAR image in Fig. 9, where the wakes of the farms at around 7.75°E are aligned in HARMONIE, but are not completely aligned in WRF. The length of the wind farm wakes of the southern wind farms are in agreement between WRF and HARMONIE, as can also be seen in the more detailed Fig 12a, where we compare the length of the wakes for Nordsee One and Gode Wind 1 and 2.

For the northern wind farms, the wake lengths is also pretty similar, but the wakes are more blurry in HARMONIE than they are in WRF. This might be due to the different grid resolution both horizontally, but also vertically, since the SAR image is as 10 m height and thus does not show the wake at hub height, but close to the surface.

*10. Figure 10, the differences between HARMONIE and WRF are large. Can this compromise the comparison between Fitch and EWP for the two models?*

This case on 08 August 2017 is the most challenging case out of the three cases due to very unsteady winds across the domain. This is also reflected in the worse scores displayed in Table 6 compared to Table 5 (see also Figure 1 in this document). Nevertheless, the general pattern at hub height is simulated similarly in WRF and HARMONIE (Fig. A2 in the manuscript). For the objective to compare FITCH and EWP for the two models, as displayed in Fig. 12, the scenario without wind farms (NWF) is used to normalize the impacts of the wind farms. Therefore, we think that the objective can still be achieved with the present set of simulations.

---

## Referee Report (RR1)

**Manuscript Review: "Modelling Wind Farm Effects in HARMONIE-AROME (Cycle 43.2.2) – Part 1: Implementation and Evaluation"**

**General**

The manuscript addresses the crucial consideration of Wind Farm Parameterization (WFP) in Numerical Weather Prediction (NWP) models, focusing on power production forecasting and the local weather impact of wind farms. The authors integrate Explicit Wake Parameterization (EWP) into the widely used HARMONIE-AROME model, comparing it with the existing FITCH wind farm model. The EWP relies on the actuator disc thrust force and incorporates wake vertical expansion within grid resolution limitations, without additional Turbulent Kinetic Energy (TKE) like the Fitch Scheme. The study rigorously compares these WFPs in HARMONIE-AROME and WRF across different settings and measurements. The paper is well-written, presenting information concisely, offering practical insights for research and application in wind energy.

**Main Point:**

A notable drawback of EWP, compared to FITCH, is the absence of the TKE source term. The paper argues for this by assuming that the heterogeneous part is a component of the mean flow; thus, an additional TKE source is not necessary. However, I contend that for turbine wakes, especially considering rotational motion and tip vortex variations that are subgrid scale and cannot be resolved by the mean flow in the mesoscale model. Furthermore, the TKE may arise from vertical shear due to the high vertical resolution; however, horizontal shear also cannot be resolved with a resolution of a few kilometers. The lack of TKE consideration affects wind profiles and wake recovery, leading to underestimation of wake effects above hub height. A more in-depth exploration of this limitation in the discussion would enhance guidance for WFP selection.

**Other Points:**

1. Equations (2-4), the core of EWP, are complex. The derivation process should be either shown or referenced to aid in understanding the underlying physical assumptions. Furthermore, it needs clarification whether these additions are new or identical to the EWP in WRF.

2. Figure 4 raises questions about Ct being larger than 1 at low ambient wind speed. An explanation in the manuscript would enhance understanding.

3. Line 197-197: "the standard deviation of errors (STDE) assesses the non-systematic error". I am not sure how useful of STDE in this paper (mentioned only once later in the paper.) or in general. For example, consider a two cases with equally large bias, where the first one has a correlation of 0 (i.e. random error+systematic error), and the second one has a

correlation of 1 (totally systematic). However, STDE for the two cases can be equal, which limits the intepretation of it.

4. Line 113: The abbreviation "IFS" needs clarification.

5. Figs. 8, 11, 13: I don't understand why the authors show two line for each simulation instead of one time-interpolated line to the measement time. The figures are also quite small to see. Some suggestion: rearrange the right legend to the top or bottom and elimitate the white space in each subplots; change color codes in to a more conistent way (e.g. red for HAR, blue for WRF, solid for FITCH, dashed for EWP, dotted for NWF).

---

## Author Response (AR3)

**Response to reviewer**

We would like to thank the reviewer for their constructive comments. We responded in detail to them below in blue.

*General*
*The manuscript addresses the crucial consideration of Wind Farm Parameterization (WFP) in Numerical Weather Prediction (NWP) models, focusing on power production forecasting and the local weather impact of wind farms. The authors integrate Explicit Wake Parameterization (EWP) into the widely used HARMONIE-AROME model, comparing it with the existing FITCH wind farm model. The EWP relies on the actuator disc thrust force and incorporates wake vertical expansion within grid resolution limitations, without additional Turbulent Kinetic Energy (TKE) like the Fitch Scheme. The study rigorously compares these WFPs in HARMONIE-AROME and WRF across different settings and measurements. The paper is well-written, presenting information concisely, offering practical insights for research and application in wind energy.*

*Main point*
*A notable drawback of EWP, compared to FITCH, is the absence of the TKE source term. The paper argues for this by assuming that the heterogeneous part is a component of the mean flow; thus, an additional TKE source is not necessary. However, I contend that for turbine wakes, especially considering rotational motion and tip vortex variations that are subgrid scale and cannot be resolved by the mean flow in the mesoscale model. Furthermore, the TKE may arise from vertical shear due to the high vertical resolution; however, horizontal shear also cannot be resolved with a resolution of a few kilometers. The lack of TKE consideration affects wind profiles and wake recovery, leading to underestimation of wake effects above hub height. A more in-depth exploration of this limitation in the discussion would enhance guidance for WFP selection.*

We fully agree that the correct parameterization of the TKE effect of wind turbines is important. In EWP wind turbines are only an implicit source of TKE due to the formation of TKE due to vertical shear in the wake wind profile. As you pointed out correctly, this study hints that an explicit TKE source might be required to fully account for the effects of the turbines. To make that clearer, we added the following sentences to the conclusion section:
"Nevertheless, this study indicates that taking only the implicit TKE formation due to vertical shear into account is not sufficient. Instead, an explicit source of TKE is required to consider the TKE formation from the rotational motion of the rotor as well as from tip vortices. Furthermore, this study showed that EWP also exhibits a different wake recovery at hub height as well as a different vertical wake profile of TKE, wind speed and other parameters. The reason for these differences are both the vertical wake expansion considered in EWP as well as the missing explicit TKE source. However, observations of the vertical profile in the wake were not available for comparison and thus further studies are necessary to investigate the correct shape of the profile in the wake."

*Other Points:*
*1. Equations (2-4), the core of EWP, are complex. The derivation process should be either shown or referenced to aid in understanding the underlying physical assumptions. Furthermore, it needs clarification whether these additions are new or identical to the EWP in WRF.*

Thank you for pointing this out. We have added the following at the beginning of section 2.1 "The theory and derivation of EWP are described in detail in Volker et al. 2015. For convenience we repeat the main concepts here. "

In addition, we added in the end of section 2.1 "This implementation is similar to the implementation of EWP in WRF except that turbulent diffusion coefficient, K, is different, since it is provided by a different PBL scheme. Having EWP implemented in HARMONIE allows for comparisons of different WFPs in HARMONIE as well as comparisons of WFE in HARMONIE and WRF."

*2. Figure 4 raises questions about Ct being larger than 1 at low ambient wind speed. An explanation in the manuscript would enhance understanding.*

Figure 4 shows Ct for the complete operating conditions, i.e. at very low wind speeds below 3 or 4 m/s depending on the turbine type the turbine is not operating and therefore no Ct value defined. Thus, the thrust coefficient is always smaller or equal to 1 and for low ambient wind speeds above cut-in Ct approaches 1. Hence, Ct is not larger than 1 at low ambient wind speed. To make this clearer, we have added "for the complete range of operating wind speeds" in "The thrust and power curves of these three wind turbine models for the complete range of operating wind speeds are shown in Fig. 4. Note that Ct is always smaller or equal to 1, since below 3 ms$^{-1}$ or 4 ms$^{-1}$, depending on the turbine model, the turbines are not operating and therefore Ct is not defined. The Ct curves cannot be extrapolated to lower wind speeds."

*3. Line 197-197: "the standard deviation of errors (STDE) assesses the non-systematic error". I am not sure how useful of STDE in this paper (mentioned only once later in the paper.) or in general. For example, consider a two cases with equally large bias, where the first one has a correlation of 0 (i.e. random error+systematic error), and the second one has a correlation of 1 (totally systematic). However, STDE for the two cases can be equal, which limits the intepretation of it.*

We agree with you that for the two cases that you describe the STDE could be the same. Thus, we agree that the STDE standalone is not a good metric to describe the accuracy of the model, since the interpretation is limited in that case. However, in combination with other metrics such as the BIAS and the correlation, the STDE completes the picture of the model evaluation: the BIAS assesses a systematic deviation of the simulation from the observation while the correlation coefficient assesses the strength of a linear relationship between the simulation and the observation. The STDE is similar to the correlation, but is not dimensionless and can therefore give additional insight. Thus, we think that the STDE along with the other error measures provides a useful addition.

In the manuscript, we use the STDE along with BIAS, RMSE and correlation to assess the overall simulation quality. All metrics are given in Tables 3, 5 and 6 and are therefore mentioned more than once in the manuscript, albeit not in the text. We therefore think that the STDE is a useful addition to the other metrics.

We have added the following to the manuscript to make it clearer: "CORR and STDE evaluate similar aspects of the model performance, but STDE is not dimensionless and therefore gives additional insights." and "While two different models can perform the same in terms of one error measure, e.g. the same correlation coefficient, they might perform differently in terms of another error measure, e.g. different BIASes. Thus, having four different error measures has the advantage that different aspects of the performance can be evaluated."

*4. Line 113: The abbreviation "IFS" needs clarification.*

We have changed the sentence containing IFS to: "We run HARMONIE in forecasting mode using hourly boundary fields from the Integrated Forecasting System (IFS) global model at ECMWF as lateral boundary conditions."

We also clarified the abbreviation NEA: "We simulate the Northern Europe DMI domain A (NEA)".

*5. Figs. 8, 11, 13: I don't understand why the authors show two line for each simulation instead of one time-interpolated line to the measement time. The figures are also quite small to see. Some suggestion: rearrange the right legend to the top or bottom and elimitate the white space in each subplots; change color codes in to a more conistent way (e.g. red for HAR, blue for WRF, solid for FITCH, dashed for EWP, dotted for NWF).*

We show two lines for each simulations, since the model output time step does not correspond to the exact transect time of the aircraft, since the model output time step is 10 minutes for WRF and 15 minutes for HARMONIE. In addition, the passage of the transect took the aircraft around 10 minutes during which the conditions might have changed. This is also highlighted by the two lines. For clarity we added to the manuscript: "For the models, the two closest model output time steps (Table 1) to the transect times are shown to highlight the temporal variability of the conditions during the passing of the aircraft over the transect. Thus, for a good match between model and observations, the observations should be within the shaded area of the model output. Overall the wind speed at 250~m matches well with the observations for some transects, even though WRF overestimated the 10~m wind speed if compared to SAR (Fig. 7)."

We moved the legend to the bottom to make the figures wider. However, the long caption of Figure 8 restricts the height of the figure and thus it cannot be enlarged further. However, the online version of the manuscript enables to zoom in and investigate dome details.

Thank you for your suggestion for a different color scheme scheme. However, we feel that we are already using a consistent color- and line-style coding with brighter colours and more dense broken lines for WRF and darker colours and more loose broken lines for HARMONIE, as described in the caption of Figure 8: "WRF simulations (brighter colours, densely broken lines) and HARMONIE simulations (darker colours, loosely broken lines)". EWP is always shown by dotted lines, FITCH by dashed lines and NWF by solid or dashed-dotted lines for WRF and HARMONIE, respectively. Having only two colours and the same line-style would make it more difficult to distinguish the model outputs, especially for color-blind people.

We modified Figure 8, 11 and 13 as well as A1 and A2 to make the figures consistent.